# Focus Your Attention when Few-Shot Classification

**Haoqing Wang**     **Shibo Jie**     **Zhi-Hong Deng**[*]
School of Intelligence Science and Technology, Peking University
{wanghaoqing,   parsley,   zhdeng}@pku.edu.cn

## Abstract

Since many pre-trained vision transformers emerge and provide strong representation for various downstream tasks, we aim to adapt them to few-shot image classification tasks in this work. The input images typically contain multiple entities. The model may not focus on the class-related entities for the current few-shot task, even with fine-tuning on support samples, and the noise information from the class-independent entities harms performance. To this end, we first propose a method that uses the attention and gradient information to automatically locate the positions of key entities in the support images, denoted as *position prompts*. Then we employ the cross-entropy loss between their many-hot presentation and the attention logits to optimize the model to focus its attention on the key entities during fine-tuning. This ability then can generalize to the query samples. Our method is applicable to different vision transformers (e.g., columnar or pyramidal ones), and also to different pre-training ways (e.g., single-modal or vision-language pre-training). Extensive experiments show that our method can improve the performance of full or parameter-efficient fine-tuning methods on few-shot tasks. Code is available at `https://github.com/Haoqing-Wang/FORT`.

## 1   Introduction

Based on the accumulated experience, humans can learn from few labeled samples for solving novel tasks, and this is a key aspect of human intelligence. Few-shot learning [12, 31] aims to imitate this ability and has attracted widely attention from the machine learning community [71]. Traditionally, a base dataset is collected for learning prior knowledge before solving novel few-shot tasks. The meta-learning methods learn the task-shared inductive bias, e.g., parameter initialization [13, 41], metric function [52, 32, 54] or optimizer [2, 47]. The transfer learning methods [56, 48] learn generalizing representation to novel tasks, and achieve comparable or better performance.

Some works [56, 46] find that the well-learned embedding model contributes the most to few-shot classification performance, and even more important than the sophisticated meta-learning algorithms. However, the commonly-used base datasets are far from large enough for learning generalizing embedding, e.g., miniImageNet [60], which severely limits the performance of few-shot classification. In fact, the few-shot learning ability of humans benefits from the supervised or unsupervised learning from *massive* signals since birth, which helps us accumulating enough general knowledge. On the other hand, self-supervised learning [66, 19, 8, 18, 5] can learn well-generalizing representation from nearly-free massive unlabeled data and is beneficial for broad downstream tasks. Using these pre-trained models for few-shot tasks conforms to the learning paradigm of humans and has achieved remarkable success [5, 74, 15] in Natural Language Processing (NLP). In this work, we aim to adapt the pre-trained models to few-shot image classification tasks, without using any base datasets.

Recently, vision transformers [11, 36, 70] play an increasingly important role in computer vision, so we use them as the backbone for few-shot tasks. Note that it is a non-trivial problem to fine-tune the pre-trained large vision transformers on few support samples due to their data-hungry

---

[*]Corresponding author.

37th Conference on Neural Information Processing Systems (NeurIPS 2023).

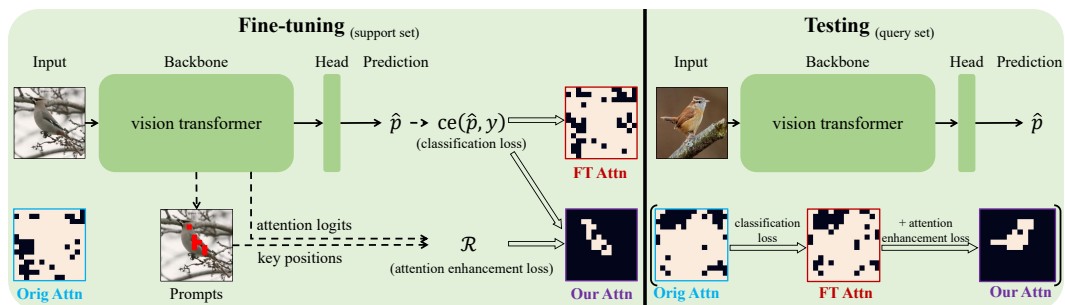

Figure 1: Focusing on key entities via position prompts. The original pre-trained model may attend to multiple entities in a single image, and the information from class-independent entities is actually noise in the current task. The extremely few support samples make the standard fine-tuning with only classification loss unable to remove the noise information. We propose position prompts (red patches) to prompt the model where are the key patches of the input and focusing on them. This ability gained during fine-tuning can generalize from support samples to query ones. Here the white patches in attention visualization have the top highest attention scores and cover about $95\%$ attention.

property. Inspired by the prompt methods [72, 35] from NLP, some works [78, 77, 10] synthesize the classification weights by encoding the learnable text prompts that describe the objective classes, and then compare them with image features. Despite impressive performance on few-shot tasks, these works strongly rely on the semantic alignment between image encoder and text encoder, and thus are specific to vision-language pre-trained models [45]. In this work, we design a new form of prompts for vision data, named *position prompts*, to adapt the pre-trained vision transformers to few-shot image classification tasks. They can prompt explicit valuable information, and also be usable to different pre-trained vision transformers, without limited structures or pre-training ways.

Figure 1 illustrates our framework and motivation, where we visualize the patches with top highest attention scores and covering about $95\%$ attention (details at Section 4.2) of [CLS] token in a ViT-B/16 [11]. The input images usually contain several different entities, e.g., a *bird* stands on the *branch* (Figure 1, 'Input'). The pre-trained model may attend to multiple entities at the same time (Figure 1, 'Orig Attn') and the noise information from class-independent entities harms performance. The standard fine-tuning methods tune full or partial [24, 34] parameters of the model using the classification loss on the support samples, and can achieve better performance than the backbone-frozen methods. But even with fine-tuning, the extremely few support samples prevent the model from giving sufficient attention to the *class-related* key entities (Figure 1, 'FT Attn'). To this end, besides the classification loss, we use the position information of the key entities in each support sample as guidance for fine-tuning, and the model is expected to focus its attention on the key entities after fine-tuning, not only for the support samples, but also for the query ones. Concretely, we first automatically locate the position of key patches as position prompts (Figure 1, 'Prompts') in each support sample based on the deep explanation method, where the key patches refer to the most *class-related* ones and correspond to key entities. Inspired by Rollout [1], we integrate the attention information from multiple layers linearly for explanation, and further introduce gradient information as an auxiliary to ensure class-specific explanation. Unlike existing prompts [35], ours are only used to guide the fine-tuning process and not available during testing, since we need label information to obtain them. Then, we use the position prompts as the prediction target of attention, and optimize the cross-entropy loss between their many-hot presentation and the attention logits during fine-tuning. This loss enables the model to identify the key patches and focus on them (Figure 1, 'Our Attn'), and this ability obtained during fine-tuning can generalize to query samples. We also theoretically prove that our method can increase the information from the key patches in the output feature and reduce that from other patches (noise information).

Overall, our contributions can be summarized as follows:

- We propose a new form of prompts for vision data, position prompts, to effectively adapt the pre-trained vision transformers to few-shot image classification tasks.
- Our method, denoted as FORT (FOcus youR aTtention), is applicable to the attention modules in different vision transformers, e.g., Swin [36] or ViT, and not limited to specific pre-training ways, e.g., single or multi-modal ones.

- We conduct extensive experiments and verify that our method can effectively improve the performance of full or parameter-efficient [24, 34] fine-tuning methods on few-shot tasks.

## 2    Related work

**Few-shot classification.**    The traditional paradigm of few-shot classification is the meta-learning framework [22, 67], which learns the task-shared inductive bias [13, 52, 32, 63] on the training tasks constructed from a base dataset. When the base dataset is unlabeled or in the different domain to novel tasks, we also need to solve new challenges, unsupervised [23, 28, 64] or cross-domain [58, 62, 14] few-shot classification. Some works [56, 46] find the well-generalizing representation is actually the key factor of few-shot ability, while the base dataset is typically not large enough to obtain it. In fact, adapting self-supervised pre-trained models [6, 76, 65] to few-shot tasks can address all these concerns simultaneously: the training data is unlabeled and thus can be sufficiently large, and the novel tasks can come from different target domains. The pre-trained large models have far more generalizing representation and thus can obtain significantly better performance than the traditional setting, as shown in Appendix B. Besides, it is more friendly to privacy data, and we only need the pre-trained models instead of pre-training data. It also conforms to the learning paradigm of humans whose few-shot ability comes from learning on massive signals since birth, and has better realistic value. An exploratory work [9] uses the ensemble of frozen pre-trained models and a simple classifier, but without further adaptation to novel tasks. CLIP [45] introduces the vision-language pre-trained models, and some works [78, 77, 10] use the text prompts to solve few-shot image classification based on them. These methods are limited to specific pre-trained models, and typically require the test classes to be seen in the training corpus [40]. Humans actually can recognize unseen objects based on only reference images, without knowing their class names. Our method is usable for both single and multi-modal pre-trained models. Similar to our work, some existing models [69, 79] also propose to highlight the key local features of each input for few-shot classification. However, they all design parametric modules and meta-learn them using the base dataset. It is not suitable for our new setting, since the parametric modules can not been learned effectively using only few support samples. Our method instead introduces no new parametric modules and does not need meta-training.

**Prompt engineering.**    Designing prompts [35] for adapting the pre-trained models to few-shot tasks can originate from GPT-3 [5], which uses language prompts for in-context learning. Since the manual prompts require domain expertise and are likely to be sub-optimal even with significant effort, many works [15, 72, 29] focus on automating prompt generation to fine-tune pre-trained language models. The language prompts can also be used for few-shot image classification [78, 77, 7] based on special vision-language pre-trained models [45]. VPT [26] introduces learnable prompts to the input space of each transformer layer, and is suitable for both single and multi-modal pre-trained models. But VPT performs unsatisfactorily on few-shot tasks, even worse than simple plug-and-play solvers [4, 32] with frozen backbone (Table 1). Its prompts do not prompt any explicit information and still follow the form of language prompts. In this work, we propose position prompts for vision data to explicitly prompt the key positions of input images to improve fine-tuning performance.

**Deep explanation method.**    In order to build trust in intelligent systems, deep explanation methods [16] calculate the local relevancy of the input with model predictions, and can be used to locate the key parts. Many classic works have been designed for CNNs, such as gradient based methods [51, 50, 55, 53] and relevance propagation based methods [3, 38, 25, 17]. For Transformers, these methods can not take advantage of the core attention information. We can use the raw attention scores to explain model predictions, but this ignores the attention components of other layers. Rollout [1] thus rolls out the attention scores of multiple layers to capture the propagation of information from input, but it is not class-specific. Inspired by Rollout, we also integrate the attention information from multiple layers, and further introduce gradient information to ensure class-specific explanation.

## 3    Model

### 3.1    Problem formulation

In this work, we aim to adapt the pre-trained vision transformers to few-shot image classification tasks. These backbones can be columnar or pyramidal architectures, e.g., ViT [11] or Swin [36], and

are pre-trained on massive data. In a few-shot classification task, we are typically given a support set $\mathcal{T}_s$ as reference and a query set $\mathcal{T}_q$ for test. The support set $\mathcal{T}_s = \{(x_i^s, y_i^s)\}_{i=1}^{C \times K}$ contains $C$ different classes with $K$ labeled samples in each class, which is denoted as '$C$-way $K$-shot'. $K$ is typically very small, e.g., 1 or 5, which makes it a challenging task, especially for the data-hungry vision transformers. The query set $\mathcal{T}_q = \{(x_j^q, y_j^q)\}_{j=1}^M$ contains the samples belong to the same classes with the support set, and we need to classify them correctly. Adapting a pre-trained model $f_\theta$ to a few-shot task $\mathcal{T} = \{\mathcal{T}_s, \mathcal{T}_q\}$ can be formulated as

$$\psi = \mathcal{A}(f_\theta, \mathcal{T}_s) \tag{1}$$

where the task solver $\mathcal{A}$ is designed to obtain the solution model $\psi$ of task $\mathcal{T}$. For example, the task solver $\mathcal{A}$ can be linear probing [9] that combines the frozen model $f_\theta$ with a linear classifier, or the parameter-efficient fine-tuning methods [34, 24, 27] that only tune partial existing or newly-added parameters for easy deployment, or using language prompts to generate classifier weights when $f_\theta$ is a vision-language pre-trained model [78, 77, 7]. The performance of solution $\psi$ is evaluated on the query set $\mathcal{T}_q$ with accuracy and so on. In this work, we aim to design better task solver $\mathcal{A}$.

## 3.2 Multi-head self-attention

The Multi-head Self-Attention (MSA) modules are the core components of various vision transformers. The input image is split into multiple patches, which are then projected to patch tokens. The learnable [CLS] token is appended to these tokens for final classification if needed. Let the input of a MSA module be $\mathbf{Z}_{in} \in \mathbb{R}^{N \times d}$, containing $N$ tokens with dimension $d$. Here $\mathbf{Z}_{in}$ may contain all input tokens for global attention [11], or partial ones for window attention [36]. MSA uses $H$ attention heads to jointly attend to the information from different representation sub-spaces. Within $h$-th attention head, we first project $\mathbf{Z}_{in}$ to Query $\mathbf{Q}_h$, Key $\mathbf{K}_h$ and Value $\mathbf{V}_h$ as

$$\mathbf{Q}_h = \mathbf{Z}_{in}\mathbf{W}_q^h \in \mathbb{R}^{N \times d_k} \qquad \mathbf{K}_h = \mathbf{Z}_{in}\mathbf{W}_k^h \in \mathbb{R}^{N \times d_k} \qquad \mathbf{V}_h = \mathbf{Z}_{in}\mathbf{W}_v^h \in \mathbb{R}^{N \times d_v} \tag{2}$$

where $\mathbf{W}_q^h$, $\mathbf{W}_k^h$ and $\mathbf{W}_v^h$ are projection matrices. The self-attention matrix $\mathbf{A}_h$ is calculated as

$$\mathbf{A}_h = \text{softmax}(\mathbf{Q}_h\mathbf{K}_h^T / \sqrt{d_k}) \in \mathbb{R}^{N \times N} \tag{3}$$

The matrix $\mathbf{A}_h$ reflects the attention of different tokens to each other, and the large score $\mathbf{A}_h^{ij}$ indicates the $i$-th token strongly attends to the $j$-th token. The output feature of the MSA module is

$$\text{MSA}(\mathbf{Z}_{in}) = \text{Concat}([\mathbf{A}_h\mathbf{V}_h]_{h=1}^H)\mathbf{W}_o + \mathbf{Z}_{in} \tag{4}$$

where $\text{Concat}(\cdot)$ denotes the concatenation operation, $\mathbf{W}_o$ is a projection matrix. Obviously, the MSA module learns the increment of each token and introduces the information from other tokens, whose quantity is controlled by the attention matrices $\{\mathbf{A}_h\}_{h=1}^H$.

## 3.3 Analysis and motivation

We first examine various designs of the task solver $\mathcal{A}$: 1) simple machine learning classifiers, e.g., Nearest Neighbor classifier, Ridge Regression and Support Vector Machines; 2) plug-and-play inductive meta-solvers (e.g., ProtoNet [52], R2D2 [4] and MetaOptNet [32]) and linear probing; 3) full or parameter-efficient fine-tuning, e.g., VPT [26], LoRA [24] and SSF [34]. The results are provided in Table 1, where we use the ViT-B/16 pre-trained on ImageNet-1K [49] with DINO [6] as $f_\theta$ and conduct on CUB [61], Cars [30], Places [75] and Plantae [59] datasets. For each fine-tuning method, we use the best linear meta-solver, MetaOptNet, to obtain the initialization of the classifier weights. Otherwise, these fine-tuning methods perform even worse than MetaOptNet on few-shot tasks (Figure 2). As shown in Table 1, the fine-tuning methods can achieve better performance than the backbone-frozen methods in most cases. But they sometimes cannot significantly improve performance beyond the classifier initialization (i.e., MetaOptNet) or even perform worse, especially on 1-shot tasks. In fact, the main challenges of few-shot tasks are: 1) the classes in the novel tasks could be unseen

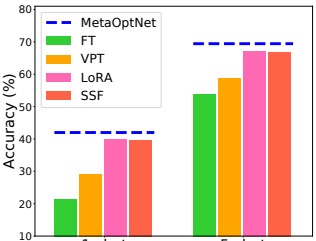

Figure 2: Average 20-way accuracy on CUB, Cars, Places and Plantae datasets. Here all fine-tuning methods use the randomly-initialized classifier.

during pre-training phase; 2) there are only few ($1 \sim 5$) labeled support samples for each class in novel tasks. The input images typically contain multiple entities. Although the pre-trained vision transformers may be able to separate different entities [6, 76], they do not pay enough attention to the key entities corresponding to the classes in the current few-shot task. Especially in the face of unseen classes, they may attend to multiple entities simultaneously [21]. When the support samples are sufficient, the model can attend to the frequently-occurring key entities in each class more to alleviate this problem, as shown in Appendix B. But the extremely few labeled samples prevent the fine-tuning methods from achieving this purpose, and the noise information from the class-independent entities misleads the fine-tuning process and harms the performance. For qualitative evidence, we visualize the patches with top highest attention scores and covering about $95\%$ attention of [CLS] token in Figure 4, more details and quantitative evidence at Section 4.2. As we can see, both the original model and fine-tuned one can not focus most attention on the key entities. To this end, we propose the position prompts to explicit prompt the model where are the key entities and guide the model to focus on them during fine-tuning. This ability then can generalize to the query samples.

### 3.4 Position prompts

We need to locate the positions of key entities in each support image as prompts for fine-tuning, i.e., the most class-related key patches for the current few-shot task. Manual labeling may provide accurate positions, but requires domain expertise and labor costs. We aim to design an automatic method, which should 1) be class-specific for locating class-related patches, and 2) make full use of the attention information, the core content of Transformers. Deep explanation methods [16] calculate the local relevancy of the input with model predictions, and can be used for highlighting the key patches for classification. For Transformers, Rollout [1] finds that using the attention of top layer for explanation ignores most of the attention components, thus miscalculating the importance of the patches, so we integrate the attention information from multiple layers like Rollout. Furthermore, only using the attention information cannot achieve class-specific explanation for some pre-trained models (Figure 3), so we further introduce the gradient information as an auxiliary.

For a columnar architecture [11] with $L$ layers, let the input feature map of $l$-th layer be $\mathbf{Z}_{in}^l \in \mathbb{R}^{N \times d}$ and the attention matrices be $\{\mathbf{A}_h^l\}_{h=1}^H$. With the well-initialized classification head above, for a support sample-label pair $(x^s, y^s)$, we first compute the gradient of the prediction score for class $y^s$, $p_{y^s}$, with respect to feature map $\mathbf{Z}_{in}^l$ as $\nabla_l = \partial p_{y^s}/\partial \mathbf{Z}_{in}^l \in \mathbb{R}^{N \times d}, l = 1, \ldots, L$. The gradient computed for single sample contains many noises, which could be amplified layer by layer along the back-propagation path. To this end, we only use the gradient to the input feature of the top layer, i.e., $\nabla_L$, and reserve its first principle component for denoise. The obtained gradient term is

$$\mathbf{G} = \nabla_L \cdot \mathbf{V}_1 \in \mathbb{R}^{N \times 1}, \quad \mathbf{U}, \mathbf{S}, \mathbf{V} = \mathrm{svd}(\nabla_L) \tag{5}$$

where $\mathrm{svd}(\cdot)$ denotes Singular Value Decomposition (SVD) operation with $\nabla_L = \mathbf{U}\mathbf{S}\mathbf{V}^T$, and $\mathbf{V}_1$ is the first column vector of matrix $\mathbf{V}$, corresponding to the largest singular value of $\nabla_L$. The gradient term $\mathbf{G}$ can provide class-specific information. As shown in Equation 4, the residual connection plays a important role in propagating information from inputs to model predictions and can preserve the position information during forward propagation. To this end, we use an identity matrix $\mathbf{I}$ to augment the attention matrix. Overall, the final attention graph of $l$-th layer is

$$\widehat{\mathbf{A}}^l = \mathrm{norm}\left(\mathbf{I} + \mathbf{A}^l + \lambda \cdot \mathbf{G}^T\right) \in \mathbb{R}^{N \times N} \tag{6}$$

where $\mathrm{norm}(\cdot)$ means to re-normalize the input so that the row sum is equal to 1, $\lambda$ can control the proportion of attention and gradient information, and we average the attention matrix over all heads, i.e., $\mathbf{A}^l = \frac{1}{H}\sum_{h=1}^H \mathbf{A}_h^l$. We assume the attentions are combined linearly and consider the path along the pairwise attention graph [1]. The patch importance scores are calculated as

$$\mathbf{s} = \mathrm{mean}(\widehat{\mathbf{A}}^1 \cdot \widehat{\mathbf{A}}^2 \cdot \ldots \cdot \widehat{\mathbf{A}}^L) \in \mathbb{R}^N \tag{7}$$

where $(\cdot)$ denotes matrix multiplication, $\mathrm{mean}(\cdot)$ denotes to average the row vectors of a matrix. We use the positions of the top $P$ patches with the highest importance scores as prompts.

For a pyramidal architecture [36, 70], since the downsampling operation destroys the patch position correspondence between layers, so we directly use the attention scores average on patches as the importance scores and introduce gradient information to achieve class-specific if necessary.

Table 1: Few-shot classification accuracy (%) on 20-way 1-shot / 5-shot tasks. We use the ViT-B/16 pre-trained on ImageNet-1K [49] with DINO for backbone initialization. 'NN', 'RR' and 'FT' denotes Nearest Neighbor classifier, Ridge Regression and full fine-tuning respectively.

| Model | CUB | | Cars | | Places | | Plantae | |
|---|---|---|---|---|---|---|---|---|
| | 1-shot | 5-shot | 1-shot | 5-shot | 1-shot | 5-shot | 1-shot | 5-shot |
| NN | 52.8 | 73.0 | 20.9 | 36.2 | 49.5 | 64.3 | 35.0 | 54.3 |
| RR | 56.6 | 84.0 | 24.0 | 56.2 | 49.9 | 68.7 | 36.8 | 62.0 |
| SVM | 52.8 | 78.7 | 20.9 | 37.3 | 49.5 | 70.8 | 35.0 | 57.7 |
| ProtoNet [52] | 52.8 | 79.8 | 20.9 | 39.3 | 49.5 | 71.2 | 35.0 | 57.7 |
| R2D2 [4] | 56.7 | 84.3 | 23.8 | 56.6 | 49.7 | 68.9 | 36.8 | 62.3 |
| MetaOptNet [32] | 57.0 | 85.1 | 24.1 | 57.9 | 50.0 | 71.1 | 36.9 | 63.7 |
| Linear Probing | 41.9 | 78.2 | 18.3 | 47.2 | 41.0 | 65.8 | 27.2 | 56.6 |
| VPT [26] | 52.9 | 81.1 | 23.3 | 54.5 | 48.0 | 69.6 | 33.9 | 60.2 |
| FT | 58.0 | 88.1 | 24.1 | 66.9 | 50.3 | 72.1 | 37.0 | 66.2 |
| LoRA [24] | 57.9 | 88.2 | 23.3 | 64.3 | 49.9 | 71.3 | 37.1 | 65.7 |
| SSF [34] | 57.8 | 88.4 | 23.8 | 62.3 | 50.2 | 73.4 | 37.2 | 66.0 |
| FT + FORT | 59.5 (1.5) | 89.2 (1.1) | 25.5 (1.4) | 68.0 (1.1) | 51.1 (0.8) | 72.9 (0.8) | 38.7 (1.7) | 67.2 (1.0) |
| LoRA + FORT | 62.5 (4.6) | 89.5 (1.3) | 26.8 (3.5) | 65.7 (1.4) | 50.8 (0.9) | 72.4 (1.1) | 38.5 (1.4) | 66.9 (1.2) |
| SSF + FORT | 62.3 (4.5) | 89.6 (1.2) | 26.5 (2.7) | 64.2 (1.9) | 51.3 (1.1) | 74.4 (1.0) | 39.0 (1.8) | 67.5 (1.5) |

## 3.5 Attention enhancement

The key patches contain the most class-related information of the input images, and the model should focus its attention on them to obtain discriminative representation in the current task, not only for the support samples, but also for the query ones. Since the position prompts (or key patches) of the query samples are unknown without label information, unlike the existing prompts [35], our position prompts can not be used in the input or middle phase of forward inference. To this end, we use the position prompts as the prediction target for attention, and optimize the cross-entropy loss between their many-hot presentation[2] and attention logits during fine-tuning. As shown in Equation 2 and 3, given the input tokens (e.g., [CLS] token and patch tokens) of the top layer $\mathbf{Z}_{in}^L = \{\mathbf{z}_n\}_{n=1}^N \in \mathbb{R}^{N \times d}$, we obtain the Query $\mathbf{Q} = \{\mathbf{q}_n\}_{n=1}^N = \mathbf{Z}_{in}^L \mathbf{W}_q$ and Key $\mathbf{K} = \{\mathbf{k}_n\}_{n=1}^N = \mathbf{Z}_{in}^L \mathbf{W}_k$ for calculating the attention logits $\{\mathbf{q}_i \mathbf{k}_j^T / \sqrt{d_k}\}, i, j = 1, \ldots, N$, where $\mathbf{W}_q = \text{Concat}([\mathbf{W}_q^h]_{h=1}^H)$ and $\mathbf{W}_k = \text{Concat}([\mathbf{W}_k^h]_{h=1}^H)$. The fine-tuning objective is

$$\min_\theta \sum_{(x^s, y^s) \in \mathcal{T}_s} [\text{ce}(f_\theta(x^s), y^s) - \alpha \cdot \mathcal{R}], \quad \mathcal{R} = \frac{1}{N \cdot |\Omega|} \sum_{n=1}^N \sum_{t \in \Omega} \ln \frac{\exp(\mathbf{q}_n \mathbf{k}_t^T / \tau)}{\sum_{m=1}^N \exp(\mathbf{q}_n \mathbf{k}_m^T / \tau)} \quad (8)$$

where $\theta$ denotes tunable full or partial parameters of the model, $\text{ce}(\cdot, \cdot)$ denotes cross-entropy classification loss, $\alpha$ is a loss coefficient, $\Omega$ is the index set of the key patches in $x^s$, $\tau$ is a temperature coefficient. We calculate the objective term $\mathcal{R}$ only at the top layer to operate the features with more refine semantic. Specially, for the pyramidal architectures, $\mathcal{R}$ is applied to the top layer of each stage. This loss enables the model to automatically identify the key patches and focus on them during fine-tuning, and this ability then can generalize to the query samples. The position prompts are fixed during fine-tuning since updating them only obtains trivial improvement but introduces obvious cost.

**Theoretical analysis.** The key patches contain the most class-related information of the input and we want to increase their information in the output feature. We first prove that the objective term $\mathcal{R}$ can increase the information from the key patches in the input tokens $\mathbf{Z}_{in}^L = \{\mathbf{z}_n\}_{n=1}^N$. Concretely, we consider the mutual information $I(z, z_\Omega)$, where the random variable $z$ presents the input token with $\{\mathbf{z}_n\}_{n=1}^N$ as its samples, the random variable $z_\Omega$ presents the key patch with $\{\mathbf{z}_t | t \in \Omega\}$ as its samples. The InfoNCE lower bound estimate [42, 44] of $I(z, z_\Omega)$ thus is

$$\hat{I}_{InfoNCE}(z, z_\Omega) = \frac{1}{N \cdot |\Omega|} \sum_{n=1}^N \sum_{t \in \Omega} \left[ \ln \frac{f(\mathbf{z}_n, \mathbf{z}_t)}{\frac{1}{N} \sum_{m=1}^N f(\mathbf{z}_n, \mathbf{z}_m)} \right] \leq I(z, z_\Omega) \quad (9)$$

where the critic $f(\cdot, \cdot)$ is used to model the density ratio and can be any positive real score function. When we use a log-bilinear function as critic, i.e.,

$$f(\mathbf{z}_n, \mathbf{z}_m) = \exp(\mathbf{z}_n \mathbf{W}_q \mathbf{W}_k^T \mathbf{z}_m^T / \tau) = \exp(\mathbf{q}_n \mathbf{k}_m^T / \tau), \quad \forall n, m \in \{1, \ldots, N\} \quad (10)$$

---
[2]For index set $\Omega$, it is a vector of length $|\Omega|$, with the values of the positions in $\Omega$ be 1 and other ones be 0.

Table 2: Few-shot classification accuracy (%) on 20-way 1-shot / 5-shot tasks. We use the Swin-T/7 pre-trained on ImageNet-1k with iBOT for backbone initialization.

| Model | CUB | | Cars | | Places | | Plantae | |
|---|---|---|---|---|---|---|---|---|
| | 1-shot | 5-shot | 1-shot | 5-shot | 1-shot | 5-shot | 1-shot | 5-shot |
| MetaOptNet [32] | 49.1 | 81.1 | 21.0 | 51.1 | 48.9 | 70.4 | 36.0 | 63.1 |
| FT | 52.7 | 87.1 | 20.9 | 59.7 | 48.1 | 69.3 | 35.9 | 65.9 |
| LoRA [24] | 53.6 | 87.4 | 20.4 | 57.9 | 48.9 | 69.5 | 37.1 | 65.8 |
| SSF [34] | 54.9 | 87.6 | 21.4 | 57.2 | 48.7 | 70.8 | 35.9 | 64.7 |
| FT + FORT | 59.1 (6.4) | 88.3 (1.2) | 22.3 (1.4) | 61.6 (1.9) | 49.3 (1.2) | 70.4 (1.1) | 37.2 (1.3) | 66.9 (1.0) |
| LoRA + FORT | 59.9 (6.3) | 88.5 (1.1) | 21.9 (1.5) | 59.4 (1.5) | 49.7 (0.8) | 70.4 (0.9) | 38.7 (1.6) | 66.9 (1.1) |
| SSF + FORT | 60.1 (5.2) | 88.9 (1.3) | 23.4 (2.0) | 58.8 (1.6) | 49.8 (1.1) | 72.0 (1.2) | 37.6 (1.7) | 66.0 (1.3) |

Table 3: Few-shot classification accuracy (%) on 20-way 1-shot / 5-shot tasks. We use the ViT-B/16 pre-trained on WIT [45] with CLIP for backbone initialization.

| Model | CUB | | Cars | | Places | | Plantae | |
|---|---|---|---|---|---|---|---|---|
| | 1-shot | 5-shot | 1-shot | 5-shot | 1-shot | 5-shot | 1-shot | 5-shot |
| MetaOptNet [32] | 68.0 | 88.7 | 67.6 | 90.7 | 51.6 | 73.8 | 46.2 | 73.0 |
| zero-shot [45] | 84.1 | 84.1 | 88.0 | 88.0 | 76.6 | 76.6 | 61.2 | 61.2 |
| CoOp [78] | 84.4 | 90.4 | 91.3 | 94.6 | 77.3 | 81.1 | 63.8 | 76.2 |
| Tip-Adapter-F [73] | 86.9 | 92.0 | 92.2 | 95.3 | 79.8 | 82.0 | 68.3 | 79.3 |
| PLOT++ [7] | 87.4 | 92.0 | 92.2 | 95.5 | 79.9 | 82.7 | 67.7 | 78.8 |
| LoRA [24] | 86.3 | 92.6 | 92.3 | 95.8 | 79.8 | 84.1 | 67.4 | 80.0 |
| LoRA + FORT | 87.8 (1.5) | 93.8 (1.2) | 93.6 (1.3) | 97.0 (1.2) | 80.6 (0.8) | 84.9 (0.8) | 68.5 (1.1) | 81.0 (1.0) |

the objective term $\mathcal{R}$ satisfies

$$\mathcal{R} = \hat{I}_{InfoNCE}(z, z_\Omega) - \ln N \tag{11}$$

Therefore, increasing the objective term $\mathcal{R}$ can increase the mutual information between the input tokens and key patches. On the other hand, the objective term $\mathcal{R}$ can also be derived as

$$\mathcal{R} = \mathcal{R}^U + \mathcal{R}^A, \quad \mathcal{R}^U = -\frac{1}{N}\sum_{n=1}^{N}\ln\sum_{m=1}^{N}\exp\left(\mathbf{q}_n\mathbf{k}_m^T/\tau\right), \mathcal{R}^A = \frac{1}{N \cdot |\Omega|}\sum_{n=1}^{N}\sum_{t\in\Omega}\mathbf{q}_n\mathbf{k}_t^T/\tau \tag{12}$$

Increasing $\mathcal{R}^U$ reduces the similarity between input token pairs and obtains a more uniform distribution of input tokens, which thus increases the entropy $H(z)$. Increasing $\mathcal{R}^A$ aligns the input tokens with the key patches and makes them more similar, which thus decreases the conditional entropy $H(z|z_\Omega)$. Since $I(z, z_\Omega) = H(z) - H(z|z_\Omega)$, the objective term $\mathcal{R}$ can increase the mutual information $I(z, z_\Omega)$. In fact, if $p(\mathbf{z}_n|\mathbf{z}_m) \propto \exp\left(\mathbf{z}_n\mathbf{W}_q\mathbf{W}_k^T\mathbf{z}_m^T/\tau\right), n, m \in \{1, \ldots, N\}$, $\mathcal{R}^U$ and $\mathcal{R}^A$ have constant bias with the Monte-Carlo estimate of $H(z)$ and $-H(z|z_\Omega)$ respectively, see derivation in appendix. Finally, the objective term $\mathcal{R}$ increases the information from the key patches in the output feature (e.g., [CLS] token or average patch tokens) from two aspects: 1) it increases the attention scores of the output feature to the key patches, which directly increases their contribution and also reduces the noise information from other patches, as shown in Equation 4; 2) it increases the information from the key patches in the input tokens, which further transports to the output feature.

## 4 Experiments

### 4.1 Few-shot classification

**Settings.** The experiments are conducted on the few-shot benchmark from [58] (i.e., CUB [61], Cars [30], Places [75] and Plantae [59] datasets), and two other fine-gained datasets: Aircraft [39] and Pets [43]. We use the pre-trained ViT-B/16 released by DINO [6] and CLIP [45] and Swin-T/7 released by iBOT [76] for examining different architectures and pre-training ways. Here we only consider the contrastive pre-trained models [6] since their representations have better linear separability than the generative pre-trained models [18, 68] and are more suitable for few-shot classification. We evaluate both the backbone-frozen methods, e.g., machine learning classifiers (Nearest Neighbor classifier, Ridge Regression and Support Vector Machine), meta-solvers (ProtoNet [52], R2D2 [4]

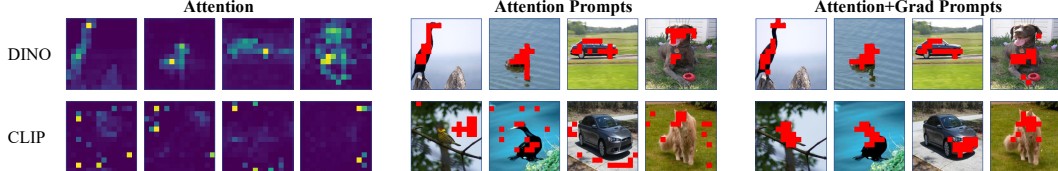

Figure 3: The attention map of [CLS] token in the pre-trained ViT-B/16 from DINO and CLIP, and the position prompts (red patches) obtained using the attention w/ or w/o gradient information.

and MetaOptNet [32]) and linear probing, and full or parameter-efficient fine-tuning, e.g., VPT [26], LoRA [24] and SSF [34]. For the vision-language pre-trained model from CLIP, we also evaluate the specially-designed models: CoOp [78], Tip-Adapter-F [73] and PLOT++ [7], and use 'a photo of a [CLASS]' as the prompts for classifier initialization in the fine-tuning methods. The language prompts are unusable for DINO and iBOT, so we use the classifier initialization based on MetaOptNet. The number of the key patches $P$ is set to 14 for ViT-B/16 and 7 for Swin-T/7. We verify the effectiveness of our model, FORT, on the fine-tuning methods. All models are evaluated in the 20-way 1-shot / 5-shot settings, instead of overly-easy 5-way settings, using 2,000 randomly sampled tasks with 15 query samples per class, and we report the average accuracy. All experiments can be conducted on single Tesla V100 with 32GB memory. More fine-tuning details are provided in Appendix B, where we also provide the results on 5-way settings for comparison with traditional setting.

Table 4: Few-shot classification accuracy (%) on 20-way 1-shot / 5-shot tasks. We use the ViT-B/16 pre-trained on ImageNet-1K with DINO for backbone initialization.

| Model | Aircraft | | Pets | |
|---|---|---|---|---|
| | 1-shot | 5-shot | 1-shot | 5-shot |
| MetaOptNet [32] | 25.5 | 55.5 | 71.9 | 89.6 |
| FT | 27.1 | 64.6 | 71.6 | 89.0 |
| LoRA [24] | 26.0 | 62.7 | 72.7 | 89.8 |
| SSF [34] | 26.5 | 61.6 | 72.7 | 90.0 |
| FT + FORT | 28.7 | 66.0 | 72.6 | 89.9 |
| LoRA + FORT | 31.3 | 63.9 | 74.8 | 90.9 |
| SSF + FORT | 30.9 | 63.1 | 74.4 | 91.2 |

Table 5: Attention-weighted summation of importance scores ($\times 10^{-2}$) on 20-way 1-shot / 5-shot tasks. We use the ViT-B/16 pre-trained on ImageNet-1K with DINO for initialization.

| Model | 1-shot | | 5-shot | |
|---|---|---|---|---|
| | support | query | support | query |
| Orig Attn | 0.72 | 0.72 | 0.72 | 0.72 |
| FT | 0.74 | 0.73 | 0.74 | 0.75 |
| LoRA [24] | 0.71 | 0.71 | 0.75 | 0.76 |
| SSF [34] | 0.74 | 0.73 | 0.79 | 0.79 |
| FT + FORT | 0.80 | 0.77 | 0.90 | 0.91 |
| LoRA + FORT | 1.00 | 0.97 | 0.96 | 0.97 |
| SSF + FORT | 1.06 | 1.03 | 1.04 | 1.04 |

**Results.** The results are provided in Table 1, 2, 3 and 4, where the 95% confidence interval is ±0.3 (1-shot) or ±0.2 (5-shot) and the number in parentheses is the increment of accuracy beyond the base fine-tuning methods. As shown in Table 1, 2 and 4, with classifier initialization, the fine-tuning methods typically can achieve better performance than the model-frozen methods, since the backbone is further adapted to task distribution. However, fine-tuning sometimes struggles to achieve non-trivial improvement beyond the classifier initialization (i.e., MetaOptNet) or even performs worse, especially on the 1-shot tasks. Our FORT can help the fine-tuning methods to obtain significant improvement on both ViT and Swin. Further, FORT typically achieves more improvement on 1-shot tasks than that on 5-shot tasks. This is intuitive, since with more support samples the model can attend to the frequently-occurring entities more in each class, which are typically the key entities. FORT is more suitable for solving the hardest 1-shot tasks. As shown in Table 3, on the vision-language pre-trained model, fine-tuning with simple classifier head (i.e., LoRA) achieves comparable or better performance than the specifically-designed methods: CoOp, Tip-Adapter-F and PLOT++. In fact, Tip-Adapter-F uses the ensemble of a linear classifier head and the similarity-based label propagation module. PLOT++ introduces multiple learnable language prompts and visual prompts for obtaining multiple classifier weights and image embeddings, and then calculates their optimal-transport distance for classification, which has huge computational burden. Our FORT can further improve the performance beyond the fine-tuning methods. Finally, FORT is compatible with the parameter-efficient fine-tuning even better than with full fine-tuning, which is convenient for industrial deployment.

## 4.2 Ablation studies

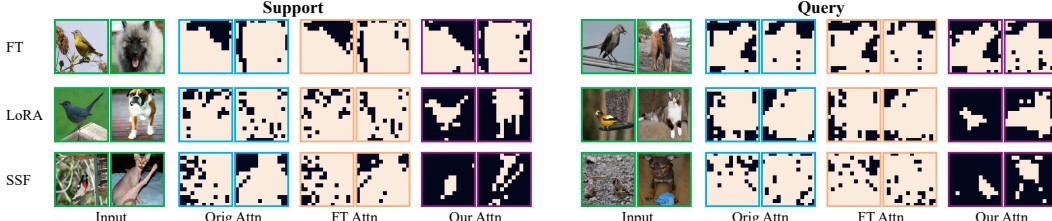

Figure 4: Visualization of the patches (white parts) with top highest attention scores and covering about 95% attention of [CLS] token in ViT-B/16. We use DINO pre-trained model for initialization.

**Position prompts.** The different pre-training ways could lead to different attention behavior in the vision transformers. Figure 3 'Attention' shows the attention map of [CLS] token in the top layer of pre-trained ViT-B/16 from both the single-modal (DINO) and multi-modal (CLIP) methods. DINO can attend to the foreground objects with the highest attention scores, while CLIP typically attends to more parts besides them. This is intuitive, since the supervision signals for the image encoder in CLIP are the sentence descriptions, which typically involve both the foreground objects and scene, e.g., 'a *bird* flies over the *sea*'. Importantly, the key entities typically can not get the highest attention scores in CLIP pre-trained model, which makes it difficult to accurately locate them only based on the attention information. Figure 3 'Attention Prompts' shows the position prompts obtained using only the attention scores, i.e., $\lambda = 0$ in Equation 6. For DINO, the obtained position prompts can indicate the key entities, while for CLIP, the obtained position prompts could point to other class-independent parts besides the key entities. To this end, we introduce the gradient information as an auxiliary. Figure 3 'Attention+Grad Prompts' shows the position prompts obtained using both the attention and gradient information, where $\lambda = 1$ for DINO and $\lambda = 50$ for CLIP. We use MetaOptNet to calculate the classifier weight for DINO, and use language prompts for CLIP. The gradient information helps to accurately locate the key entities in CLIP, and also improve that in DINO.

**Attention enhancement.** We conduct qualitative and quantitative experiments to verify that our method can enhance the attention of the model to the key patches. Qualitatively, Figure 4 visualizes the patches with top highest attention scores of [CLS] token in ViT-B/16. Concretely, when the attention scores of all patches are $a_1, \ldots, a_N$ in descending order, we choose the patches $\{1, \ldots, n\}$, which satisfy $S_{n-1} < 0.95 \leq S_n$ with $S_n = \sum_{i=1}^{n} a_i$ and $S_N = 1$. We use the pre-trained model from DINO for backbone initialization and fine-tune on random 1-shot tasks. We provide the results of original attention ('Orig Attn') and the attention after fine-tuning w/ ('Our Attn') or w/o ('FT Attn') our method. The original model can not focus its attention on the key patches, nor can the full or parameter-efficient fine-tuning achieve this property. Our method can prompt the model the positions of key patches and help the model to focus most attention (95%) on them, and this ability can generalize from the support set to query samples. This phenomenon is not obvious on full fine-tuning, and the reason may be it uses smaller learning rate and less epochs than LoRA and SSF to avoid over-fitting. The full attention maps are provided in appendix and we can observe the same phenomenon. Further, we examine the attention-weighted summation of importance scores, i.e., $\sum_{i=1}^{N} \mathbf{s}_i * \mathbf{a}_i^0$, where $\mathbf{s}$ is defined in Equation 7 and $\mathbf{a}^0$ is its position-aligned attention scores. The larger the value, the more focused on the key patches the attention. Table 5 provides the results averaged on CUB, Cars, Places and Plantae datasets with 2000 tasks each setting. The fine-tuning methods obtain the similar scores with original attention, while our FORT achieves non-trivial increase. Consistently, the increment on full fine-tuning is smaller than that on LoRA and SSF. For standard fine-tuning, the increment on 5-shot tasks is larger than that on 1-shot tasks. In fact, with more support samples, the model can attend to the frequently-occurring key entities in each class.

**Importance calculation.** In Section 3.4, we combine both gradient and attention information for calculating the importance score of each patch. Here we ablate the different information item and compare their performance. We follow the experimental settings in Section 4.1 and use the per-trained models from DINO or CLIP for initialization. The results are provided in Table 6, where 'FORT (attn)' denotes using only attention information (i.e., setting $\lambda = 0$ in Equation 6), 'FORT (grad)' denotes using only gradient term $\mathbf{G}$, 'FORT (attn+grad)' denotes using both attention and gradient information, that is our final method. Here we pick the best hyper-parameters for each method, even though they may differ from each other. For DINO pre-trained models, the attention information

Table 6: Few-shot classification accuracy (%) on 20-way 1-shot / 5-shot tasks. We use the ViT-B/16 pre-trained by DINO or CLIP for backbone initialization.

| Model | DINO | | | | CLIP | | | |
| | CUB | | Cars | | CUB | | Cars | |
| | 1-shot | 5-shot | 1-shot | 5-shot | 1-shot | 5-shot | 1-shot | 5-shot |
|---|---|---|---|---|---|---|---|---|
| LoRA | 57.9 | 88.2 | 23.3 | 64.3 | 86.3 | 92.6 | 92.3 | 95.8 |
| + FORT (attn) | 62.1 | 89.0 | 26.3 | 65.0 | 86.6 | 92.6 | 92.5 | 95.8 |
| + FORT (grad) | 60.0 | 88.8 | 25.2 | 64.7 | 87.8 | 93.8 | 93.6 | 97.0 |
| + FORT (attn+grad) | 62.5 | 89.5 | 26.8 | 65.7 | 87.8 | 93.8 | 93.6 | 97.0 |

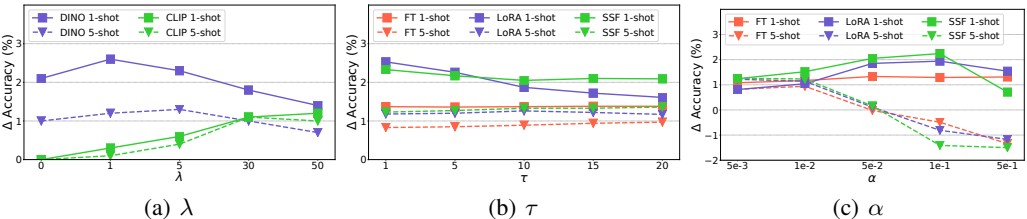

(a) $\lambda$        (b) $\tau$        (c) $\alpha$

Figure 5: Ablation studies on hyper-parameter $\lambda$, $\tau$ and $\alpha$. We examine the absolute increment of accuracy beyond the base fine-tuning methods averaged on CUB, Cars, Places and Plantae datasets.

can effectively locate the key patches and contributes more to the performance than the gradient information. For CLIP pre-trained models, the attention information can not indicate the positions of the key patches and thus the gradient information plays a major role. Combining both attention and gradient information works for different pre-trained models and can obtain the best performance.

**Hyper-parameters.** We conduct ablation studies on the hyper-parameters from Equation 6 and 8: $\lambda$, $\tau$ and $\alpha$, and examine the absolute increment of accuracy beyond the base fine-tuning methods averaged on CUB, Cars, Places and Plantae datasets. For each hyper-parameter, we set the other ones to its corresponding best values. For $\lambda$, we use LoRA as the base fine-tuning method and evaluate different pre-trained models, i.e., ViT-B/16 from DINO and CLIP. For $\tau$ and $\alpha$, we use the ViT-B/16 from DINO and evaluate different base fine-tuning methods, i.e., full fine-tuning, LoRA and SSF. Figure 5 (a) shows that using DINO pre-trained model is suitable for small $\lambda$ since its attention is enough for locating key patches, while using CLIP pre-trained model needs large $\lambda$ to utilize gradient information. Figure 5 (b) shows that our method is generally robust to the temperature coefficient $\tau$, and the performance on 1-shot tasks is relatively more sensitive than that on 5-shot tasks. Figure 5 (c) shows that the loss coefficient $\alpha$ is an important hyper-parameter and needs to be carefully chosen, and the performance on 5-shot tasks is more sensitive than that on 1-shot tasks. In fact, if the coefficient $\alpha$ is too large, the attention enhancement loss could relatively inhibit the optimization of the classification loss (e.g., skew gradient direction), which is harmful to fitting the input-label distribution. The more training samples, the more important it is to fit input-label distribution, so the 5-shot tasks prefer smaller coefficient $\alpha$ than 1-shot tasks.

## 5 Conclusion and limitations

In this work, we aim to adapt the pre-trained vision transformers to few-shot image classification tasks. It is nontrivial, since even with well classifier initialization, the standard fine-tuning methods sometimes can not obtain significant improvement beyond the initialization, especially on 1-shot tasks. To this end, we propose a new prompt for vision data, position prompts, to explicitly prompt the model where are the key entities and focus on them during fine-tuning. Our method can effectively improve the performance of full or parameter-efficient fine-tuning on few-shot tasks. The limitations of our methods are: 1) our method is suitable for multi-head attention module, and unusable for other vision backbones without attention, e.g., ResNet [20] or MLP-Mixer [57]; 2) for some professional fields, e.g., medical imaging, we may still need experts to manually label the position prompts.

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

Table 7: Candidate values of hyper-parameters for fine-tuning.

| hyper-parameter | value |
|---|---|
| learning rate | 1e-5, 3e-5, 5e-5, 8e-5, 1e-4, 3e-4, 5e-4, 8e-4, 1e-3, 3e-3, 5e-3, 8e-3, 1e-2 |
| #epoch | 5, 10, 15, 20 |
| temperature $\tau$ | 1, 5, 10, 15, 20 |
| coefficient $\alpha$ | 5e-4, 8e-4, 1e-3, 3e-3, 5e-3, 8e-3, 1e-2, 3e-2, 5e-2, 8e-2, 1e-1, 3e-1, 5e-1, 8e-1, 1 |

## A Derivation of Monte-Carlo estimate

In Section 3.5, we claim that if $p(\mathbf{z}_n|\mathbf{z}_m) \propto \exp\left(\mathbf{z}_n \mathbf{W}_q \mathbf{W}_k^T \mathbf{z}_m^T / \tau\right) = \exp\left(\mathbf{q}_n \mathbf{k}_m^T / \tau\right), n, m \in \{1, \ldots, N\}$, $\mathcal{R}^U$ and $\mathcal{R}^A$ have constant bias with the Monte-Carlo estimate of $H(z)$ and $-H(z|z_\Omega)$ respectively, where

$$\mathcal{R}^U = -\frac{1}{N} \sum_{n=1}^{N} \ln \sum_{m=1}^{N} \exp\left(\mathbf{q}_n \mathbf{k}_m^T / \tau\right), \qquad \mathcal{R}^A = \frac{1}{N \cdot |\Omega|} \sum_{n=1}^{N} \sum_{t \in \Omega} \mathbf{q}_n \mathbf{k}_t^T / \tau \qquad (13)$$

We define

$$p(z|\hat{z}) = \frac{\exp\left(z \mathbf{W}_q \mathbf{W}_k^T \hat{z}^T / \tau\right)}{Z(\hat{z})}, \qquad p(z|z_\Omega) = \frac{\exp\left(z \mathbf{W}_q \mathbf{W}_k^T z_\Omega^T / \tau\right)}{Z(z_\Omega)} \qquad (14)$$

where the random variable $\hat{z}$ follows the same distribution as $z$ and also has $\{\mathbf{z}_m\}_{m=1}^{N}$ as its some samples, and $Z(\cdot)$ is the partition function for normalization. Here we assume that the energy state remains the same for different $\hat{z}$ or $z_\Omega$, i.e., $Z(\cdot)$ keeps a constant $Z$. The detailed derivations are

$$H(z) = -\mathbb{E}_{p(z)} \ln p(z) = -\mathbb{E}_{p(z)} \ln \mathbb{E}_{p(\hat{z})} p(z|\hat{z})$$

$$\approx -\frac{1}{N} \sum_{n=1}^{N} \ln \mathbb{E}_{p(\hat{z})} p(\mathbf{z}_n|\hat{z})$$

$$\approx -\frac{1}{N} \sum_{n=1}^{N} \ln \frac{1}{N} \sum_{m=1}^{N} p(\mathbf{z}_n|\mathbf{z}_m)$$

$$\approx -\frac{1}{N} \sum_{n=1}^{N} \ln \sum_{m=1}^{N} \exp\left(\mathbf{q}_n \mathbf{k}_m^T / \tau\right) + \ln NZ$$

$$\approx \mathcal{R}^U + \ln NZ \qquad (15)$$

and

$$-H(z|z_\Omega) = \mathbb{E}_{p(z,z_\Omega)} \ln p(z|z_\Omega)$$

$$\approx \frac{1}{N \cdot |\Omega|} \sum_{n=1}^{N} \sum_{t \in \Omega} \ln p(\mathbf{z}_n|\mathbf{z}_t)$$

$$\approx \frac{1}{N \cdot |\Omega|} \sum_{n=1}^{N} \sum_{t \in \Omega} \mathbf{q}_n \mathbf{k}_t^T / \tau - \ln Z$$

$$\approx \mathcal{R}^A - \ln Z \qquad (16)$$

From these results, the above claim is proved. Besides, we have $I(z, z_\Omega) = H(z) - H(z|z_\Omega) \approx \mathcal{R}^U + \mathcal{R}^A + \ln N = \mathcal{R} + \ln N$, i.e., we obtain the Equation 11 again.

## B More experiments

**Fine-tuning details.** The resolution of input images is $224 \times 224$, and we do not use data augmentation following the standard setting. In fact, we find the base data augmentation (e.g., random resized cropping, flip or random erasing) can not obtain improvement on the chosen datasets of our

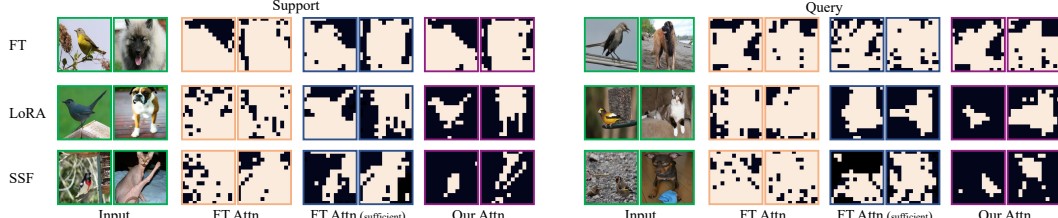

Figure 6: Visualization of the patches (white ones) with top highest attention scores and covering about 95% attention of [CLS] token in ViT-B/16. We use DINO pre-trained model for initialization.

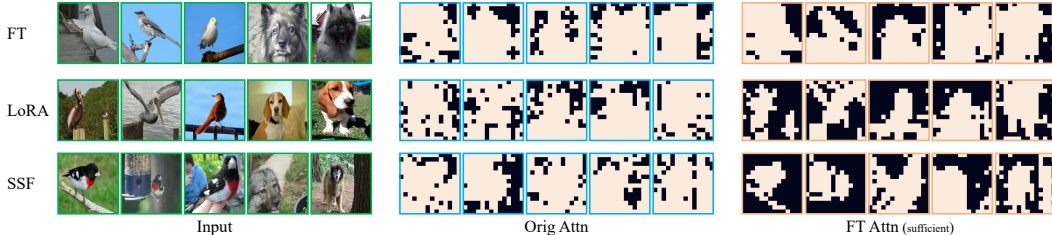

Figure 7: Visualization of the patches (white ones) with top highest attention scores and covering about 95% attention of [CLS] token in ViT-B/16. We use DINO pre-trained model for initialization.

experiments. We do not consider the pre-trained backbones from masked image modeling models for initialization, e.g., BEiT [76] or MAE [18], since these generative pre-training models can not obtain highly linearly separable features and perform poorly on few-shot tasks. We set the rank to 4 for LoRA [24] and use 10 learnable prompts at each layer for VPT [26]. We mainly use AdamW [37] optimizer for fine-tuning, except for linear probing and VPT where we use SGD optimizer with weight decay of 1e-3. We set the batch size to 20, same as the number of classes, for both 1-shot and 5-shot tasks. We set $\lambda = 1$ for DINO pre-trained model and $\lambda = 50$ for CLIP pre-trained model for simplicity. The other hyper-parameters, including learning rate, number of fine-tuning epochs, temperature $\tau$ and coefficient $\alpha$, are changeable for different fine-tuning methods, with or without classifier initialization and different datasets. To this end, we only provide their candidate values in Table 7, and choose them according to the average accuracy on 50 randomly sampled few-shot tasks. We only use 50 tasks for quick hyper-parameter selection. Considering the confidence intervals on random 50 tasks are too large to be an accurate indication of performance, e.g., $\pm 1.7$, we fix the random seed so as to compare the performance of different hyper-parameters on the same 50 few-shot tasks. We first determine the learning rate and the number of epochs for the base fine-tuning methods, and then determine the temperature $\tau$ and the coefficient $\alpha$ for our method.

**Attention maps with sufficient training samples.** We follow the experimental settings of Figure 4, except fine-tuning on the 20-way 50-shot task from CUB and 20-way 200-shot task from Pets without attention enhancement. The attention maps after fine-tuning are shown in Figure 6 and Figure 7 as "FT Attn (sufficient)". As we can see, fine-tuning with sufficient samples indeed focuses on the key entities more than with few samples ("FT Attn") or original attention ("Orig Attn"), but is less focused than explicit attention enhancement ("Our Attn"). Note that this does not mean that few-shot fine-tuning with attention enhancement can obtain higher accuracy than sufficient-shot fine-tuning. In fact, sufficient training samples allow the models to fit the input-label distribution better, which is more important for classification than attention enhancement. But when we can not obtain sufficient samples, attention enhancement can reduce class-independent noise information in the final embedding as shown experimentally and theoretically, and thus is useful for classification.

**Performance on 5-way tasks.** In order to facilitate comparison with traditional few-shot learning setting and show the advantages of our new setting, we provide the results on 5-way 1-shot / 5-shot tasks in Table 8. The performance are far more better than traditional few-shot learning setting. On miniImageNet, we can achieve 94.2% accuracy for 1-shot tasks and 98.5% accuracy for 5-shot tasks, while the SOTA traditional model, FeLMi [48], only achieve 67.47% and 86.08% accuracy for 1-shot

Table 8: Few-shot classification accuracy (%) on 5-way 1-shot / 5-shot tasks. We use the ViT-B/16 pre-trained on ImageNet-1k with DINO for backbone initialization.

| Model | miniImageNet | | CUB | | Cars | | Pets | |
|---|---|---|---|---|---|---|---|---|
| | 1-shot | 5-shot | 1-shot | 5-shot | 1-shot | 5-shot | 1-shot | 5-shot |
| LoRA | 93.1 | 97.8 | 80.1 | 94.7 | 43.3 | 74.9 | 88.7 | 95.9 |
| LoRA + FORT | 94.2 | 98.5 | 84.4 | 96.2 | 45.2 | 76.1 | 89.9 | 96.8 |
| SSF | 92.9 | 97.5 | 78.1 | 94.1 | 43.5 | 74.7 | 89.4 | 96.5 |
| SSF + FORT | 94.2 | 98.1 | 81.8 | 96.4 | 46.2 | 76.6 | 91.2 | 97.1 |

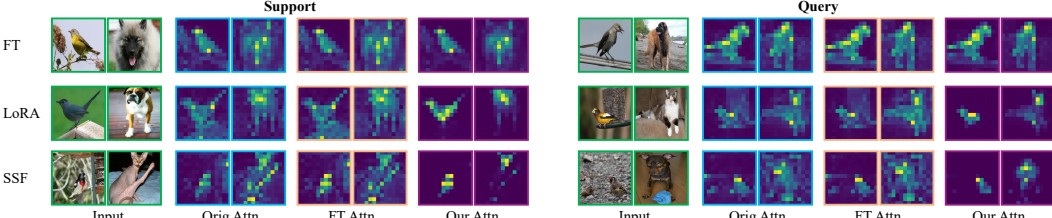

Figure 8: Visualization of the attention map of [CLS] token at the top layer of ViT-B/16. We use the ViT-B/16 pre-trained on ImageNet-1K with DINO for backbone initialization.

and 5-shot tasks respectively. Besides, although TDM [33] can achieve comparable performance on CUB with our method, it far lags behind our method on Pets, i.e., 55.8% vs 91.2% on 1-shot tasks and 72.5% vs 97.1% on 5-shot tasks. This means that TDM can not obtain stable good performance while our method can achieve satisfactory performance on different datasets and are more general. In fact, the traditional setting which uses a base dataset is not inherently a true few-shot setting. If we want to obtain a few-shot model for CUB, we have to collect many labeled bird images for meta-training, which still needs massive labeling labor and seriously hinders the practical application. Importantly, for different domains (e.g., CUB, Cars), we have to label separate base datasets for them. Our new setting uses open-source large models pre-trained on massive unlabeled data and only uses the few support samples from different target domains for fine-tuning, which is a true few-shot setting and can handle different domains simultaneously. Using pre-trained large models is more effective and practically valuable than traditional setting. Finally, our method can further improve the fine-tuning performance of base fine-tuning methods, e.g., LoRA and SSF, even on 5-way tasks.

**Attention map of different fine-tuning methods.** In Figure 4, we choose the patches with top highest attention scores and covering about 95% attention for visualization. Here we provide the full attention map of original attention ('Orig Attn') and that after fine-tuning w/ ('Our Attn') or w/o ('FT Attn') our method in Figure 8. The settings are the same, i.e., we use the ViT-B/16 pre-trained on ImageNet-1K [49] with DINO for backbone initialization and fine-tune on random 1-shot tasks. Neither the original nor fine-tuned models can focus their attention on the key entities. Our method can obviously make the model to focus on the key entities and reduces the attention to the class-independent parts. This ability can generalize from the support set to query samples.

**Fine-tuning without classifier initialization.** In Figure 2, we compare MetaOptNet [32] with full or parameter-efficient fine-tuning methods, i.e., VPT [26], LoRA [24] and SSF [34], via average accuracy on CUB [61], Cars [30], Places [75] and Plantae [59] datasets. Here we provide detailed results in Table 9, where we use the ViT-B/16 pre-trained on ImageNet-1K [49] with DINO for backbone initialization and the 95% confidence interval is ±0.3 (1-shot) or ±0.2 (5-shot). As we can see, all fine-tuning methods perform worse than the backbone-frozen MetaOptNet on few-shot tasks. The extremely few labeled samples can not simultaneously learn a good classifier and adapt the backbone to the data distribution. In fact, the randomly initialized classifier back-propagates noisy gradients, which destroys the knowledge learned during pre-training and also fails to adapt new data distributions. Further, this phenomenon also shows it is very difficult to learn new parametric modules from scratch on few-shot tasks, e.g., classifier head, and we need to construct the well-

Table 9: Few-shot classification accuracy (%) on 20-way 1-shot / 5-shot tasks. We use the ViT-B/16 pre-trained on ImageNet-1K with DINO for backbone initialization. 'FT' denotes full fine-tuning. Here all fine-tuning methods use the randomly-initialized classifier.

| Model | CUB | | Cars | | Places | | Plantae | |
|---|---|---|---|---|---|---|---|---|
| | 1-shot | 5-shot | 1-shot | 5-shot | 1-shot | 5-shot | 1-shot | 5-shot |
| MetaOptNet | 57.0 | 85.1 | 24.1 | 57.9 | 50.0 | 71.1 | 36.9 | 63.7 |
| VPT [26] | 38.3 | 73.3 | 17.5 | 43.1 | 35.9 | 64.6 | 25.1 | 54.1 |
| FT | 25.0 | 66.7 | 15.3 | 42.2 | 23.6 | 55.0 | 22.0 | 51.4 |
| LoRA [24] | 55.3 | 83.6 | 22.5 | 54.7 | 48.4 | 68.2 | 34.2 | 62.1 |
| SSF [34] | 54.8 | 83.4 | 22.6 | 53.9 | 47.9 | 69.4 | 33.7 | 61.2 |

initialization for them. Considering MetaOptNet is a linear solver with excellent performance, we use it to calculate the classifier weights as initialization, which significantly improves the performance of various fine-tuning methods. Since the prompts in VPT are learned from scratch and we can not obtain their initialization, VPT still underperforms MetaOptNet on few-shot tasks even with classifier initialization. We recommend using a linear solver (e.g., ProtoNet [52], R2D2 [4] or MetaOptNet [32]) to obtain a good initialization of the classifier as a necessary preemptive operation when adapting pre-trained models to few-shot classification tasks in the future.

**Patch attention.** In the main text, we only examine the attention of the [CLS] token at the last layer. Since our method calculates the attention enhancement loss on both the [CLS] and patch tokens, here we visualize the attention map of those tokens at the penultimate (11-th) and last (12-th) layer of ViT-B/16. We use the ViT-B/16 pre-trained on ImageNet-1K with DINO for backbone initialization, and fine-tune the model using LoRA or SSF and evaluate the effect of our method on them. We choose the [CLS] token and $\{14, 28, 42, 56, 70, 84, 98, 112, 126, 140, 154, 168, 182, 196\}$-th patches for visualization and the results are shown in Figure 9. After fine-tuning with LoRA or SSF, both the [CLS] and patch tokens still attend to the other parts besides the key entities, introducing class-independent noise information, and some patch tokens even attend to those parts more than the key entities. Our method can enhance both the [CLS] and patch attention to make them focusing on the key entities. Interestingly, although we do not calculate the attention enhancement loss at the penultimate layer, the attention of its [CLS] and patch tokens are still significantly enhanced and focus on the key entities. As the theoretical analysis in Section 3.5, our method can increase the mutual information between the input tokens of the last layer (i.e., the output tokens of the penultimate layer) and the key patches. In order to achieve this objective, the [CLS] and patch tokens of the penultimate layer should also pay more attention to the key patches to introduce more information from them.

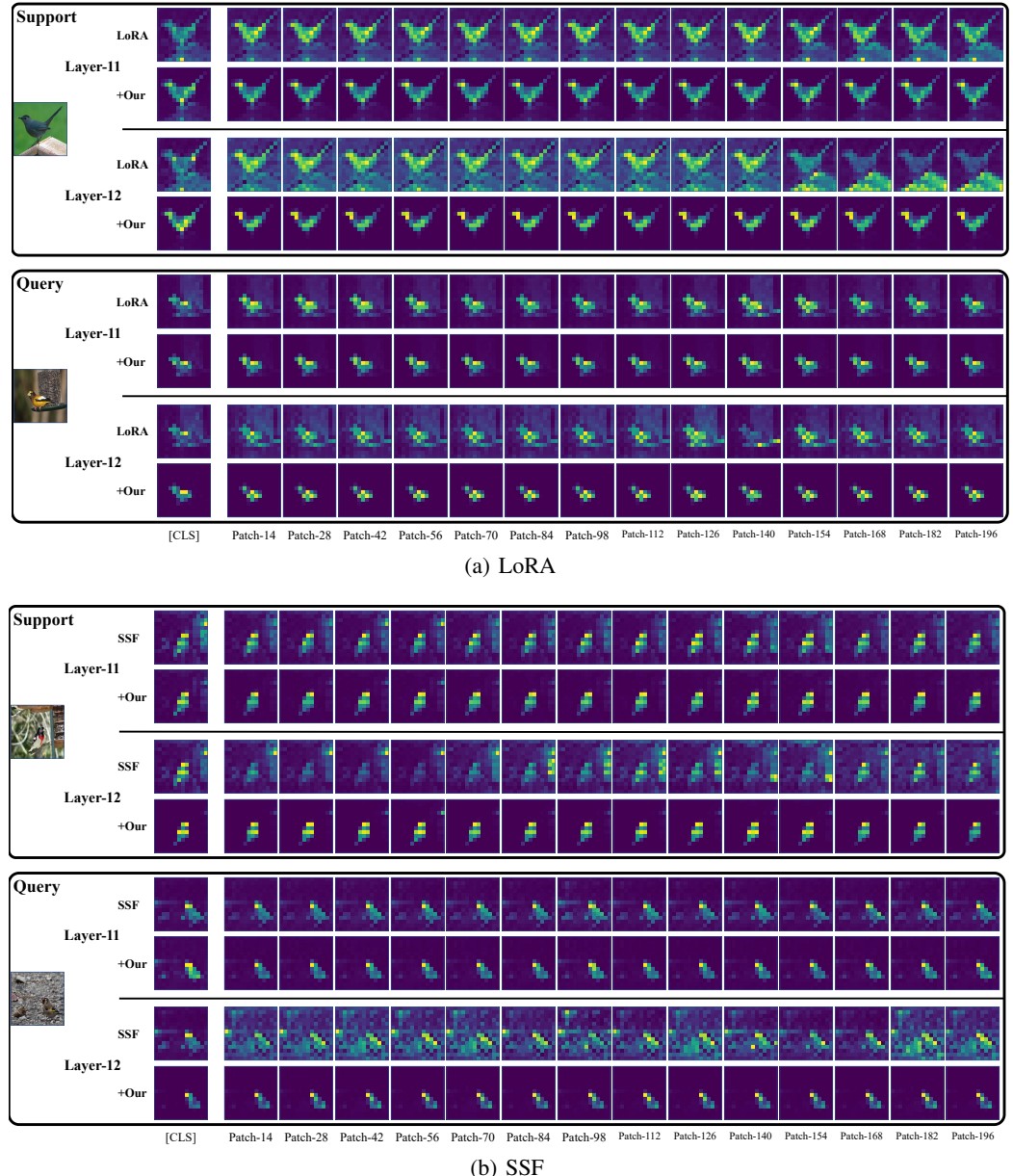

Figure 9: Visualization of the attention map of the [CLS] token and different patches at the penultimate (11-th) and last (12-th) layer of ViT-B/16. Here we use the ViT-B/16 pre-trained on ImageNet-1K with DINO for backbone initialization and evaluate the effect of our method on LoRA and SSF.

