# OpenReview forum: "Focus Your Attention when Few-Shot Classification"
_NeurIPS.cc/2023/Conference — NeurIPS 2023 poster_

### Official Review · Reviewer_nhXH · 2023-07-05

**Soundness:** 3 good
**Presentation:** 3 good
**Contribution:** 3 good
**Rating:** 5
**Confidence:** 4

**Summary:**

This paper proposes to directly adapt the large-scale pretrained model to the downstream classification task via fine-tuning on few-shot examples, therefore yielding a novel few-shot learning paradigm. Different from common few-shot classification methods, their paradigm is featured for 1) utilizing large-scale pretraining and 2) NO popular base training. Although large-scale pretraining usually promises substantial benefit and may bring unfair comparison against prior few-shot learning methods, the reviewer agrees with the author that this paradigm is more realistic and is of great value. Moreover, extensive experiments show that the proposed method achieves consistent improvement over various baselines (under the same paradigm), therefore validating the effectiveness of the proposed attention enhancement for few-shot adaptation.

**Strengths:**

1) The proposed "large-scale pretraining --> few-shot learning" (no base training) is meaningful has good realistic value.
2) Making the attention focus onto the key entities sounds reasonable for few-shot downstream task adaptation.
3) Experiments validate the method with consistent improvement over a battery of baselines.

**Weaknesses:**

- In Eqn. 6, the rationale of adding an identity matrix to the attention matrix is unclear. Is it because the attention layer contains a residual operation？
- The attention graph calculation for the columnar and pyramidal architecture differs a lot from each other.  How does the latter (and simpler) manner perform on columnar architecture? Ablation studies to investigate the detailed design of the attention graph are expected.
- The results analysis (L269 to L287) needs better clarification. Currently, multiple observations are squeezed into a single paragraph and it is not always clear which results correspond to each observation. Some analysis in the main text is not explicitly consistent with the results in the tables. For example, Line 272 states that the fine-tuning methods can achieve better performance than the model-frozen methods. This observation does not stand in Table 2, where two model-frozen methods (LoRA and SSF) are higher than FT.
- The “orig attn” visualization in Fig 1 is somehow misleading (which barely attends to the useful foreground), while the visualization in Fig. 4 is much more reasonable (the pre-trained cls token, though attends to many unrelated regions, partially covers some useful foreground). Please check whether the visualization in Fig. 1 is correct.

**Questions:**

- What is many-hot presentation？
- In Table 1 and 2, “FT” (full finetuning) is slightly higher than all the three parameter-efficient tuning methods (VPT, LoRA, SSF). This is contrary to the intuition that PET methods is superior under the few-shot learning scenario. Can you give some explanations on this phenomenon?


**Limitations:**

The authors have adequately addressed the limitations.

---

> ### Author Rebuttal · Authors · 2023-08-09
>
> Please check the reference rules in the global response first to help following reading.
>
> **1.many-hot presentation**
>
> For an input image, we obtain its position prompts, i.e., the position index set *O* of the key patches. For a vector *z* of length *N*, where *N* is the patch number, we set its values of the positions in the index set *O* to 1 and other positions to 0, then *z* is the many-hot presentation of the position prompts. Similar to one-hot label, its cross-entropy loss is actually the negative log probability at key patch positions and the expanded formula is provided in Eq.8-main. Since the attention scores are computed from Softmax, so each score can be treated as the prediction probability to the corresponding position.
>
> **2.FT vs PET in few-shot learning**
>
> For all fine-tuning methods, we initialize the classification head using closed-form linear classifier, e.g., MetaOptNet[a], before the fine-tuning process. As shown in Fig.2-main, without classification head initialization, the performance of the fine-tuning methods even lags significantly behind closed-form MetaOptNet, and PET fine-tuning indeed obtains better performance than full fine-tuning on few-shot tasks. The detailed accuracy are provided in Tab.2-appendix. The classification head initialization can significantly improve the performance of fine-tuning methods on few-shot tasks and also make full fine-tuning comparable with PET fine-tuning. We recommend this operation as the default for fine-tuning on few-shot tasks, which also is a contribution of this work.
>
> **3.Identity matrix in Eq.6-main**
>
> Yes, the rationale of adding an identity matrix to the attention matrix is the attention
> layer contains a residual connection. The residual connection plays a important role in propagating information from inputs to model predictions and can preserve the position information during forward propagation. Concretely, the input *X* and out *Y* of the attention layer approximately satisfy *Y = (W + I) X* with *W* is the attention matrix.
>
> **4.results of locating method from pyramidal architecture in columnar one**
>
> For a pyramidal architecture, since the downsampling operation destroys the patch position correspondence between layers, so we directly use the attention scores average on patches as the importance scores. The results of this simple method in columnar architecture are provided in Tab.2-rebuttal as “+ FORT (attn score)”, the method used in our paper is shown as “+ FORT”. As we can see, this simple method can not achieve as effective improvement as the used method, especially for CLIP pre-trained model whose attention is not enough to locate the key patches and we need gradient information as auxiliary.
>
> **5.analysis between Line-269 to Line-287**
>
> We guess the unclear reference to methods causes confusion. In this paper, the model-frozen methods refer to the simple machine learning classifiers (e.g., Nearest Neighbor classifier, Ridge Regression and Support Vector Machines), plug-and-play inductive meta-solvers (e.g., ProtoNet[b], R2D2[c] and MetaOptNet[a]) and linear probing. LoRA[d], SSF[e] and FT are all fine-tuning methods and adjust the parameters to adapt new tasks. Our FORT is used for these fine-tuning methods and improve their performance. Therefore, the fine-tuning methods indeed can achieve better performance than the model-frozen methods, e.g., MetaOptNet vs (SSF, LoRA or FT) in Tab.1,2,3,4-main. FORT can further help the fine-tuning methods to obtain significant improvement on both ViT and Swin, e.g., LoRA vs LoRA+FORT in Tab.1,2,4-main. We will make these more clear in the revision.
>
> **6.“orig attn” visualization in Fig.1-main**
>
> The “orig attn” visualization in Fig.1-main is correct. The original attention attends to both foreground and background objects and thus the visualization of 95% attention may cover most positions.
>
> [a]Meta-learning with differentiable convex optimization. CVPR 2019.
> [b]Prototypical networks for few-shot learning. NIPS 2017.
> [c]Meta-learning with differentiable closed-form solvers. ICLR 2019.
> [d]Lora: Low-rank adaptation of large language models. ICLR 2022.
> [e]Scaling & shifting your features: A new baseline for efficient model tuning. NeurIPS 2022.

---

> > ### Comment · Reviewer_nhXH · 2023-08-21
> > **Thanks for the rebuttal by the authors**
> >
> > The authors' rebuttal addresses most of my concern. They did not clearly explain the inconsistency between the visualization in Fig. 1 and Fig.4 (regarding the baseline attention). However, I would like to maintain my recommendation of "borderline accept" and suggest the authors give this problem a double check.

---

> > > ### Author Response · Authors · 2023-08-21
> > > **Thanks very much for you feedback.**
> > >
> > > This is a good suggestion. We carefully check the "Orig Attn" visualization in Fig.1-main and find the white (chosen) patches cover the foreground bird region. So we think it is consistent to Fig.4-main.

---

### Official Review · Reviewer_SFDV · 2023-07-06

**Soundness:** 2 fair
**Presentation:** 3 good
**Contribution:** 2 fair
**Rating:** 4
**Confidence:** 5

**Summary:**

This paper introduce a method called FORT, aiming to adapt pre-trained vision transformers to few-shot image classification task. This method contains two steps. 1) Utilizing attention and the gradient information to locate important entities, which is denoted as position prompts. 2) A new loss is defined to enforce the attention close to the position prompts. Extensive experiments shows benefits of FORT over different datasets and different pre-trained model.

Contributions can be summarized as follow:
The position prompt and the new loss can make fine-tunning on pre-trained transformer based model more efficient when dealing with few-shot samples for classification task.


**Strengths:**

1) The writing and presentation is logical.
2) The intuition to make the attention concentrate to the class related entities are reasonable.
3) Derivation of analysis makes sense.
4) Extensive experiments shows some benefit of the proposed method.

**Weaknesses:**

1) The idea of denoising the attention more interpretable seems not very novel, especially for papers of interpretability and visualizations. Would be nice to discuss more in the related work about these methods.
2) From my understanding, this method is only propose a regularizer in equation 8 combined with other standard fine-tunning procedure. The contribution of this is not significant. The improvement 1percent accuracy of experiments doesn't seem to be very siginificant.

**Questions:**

1) for section 3.4, are the position prompts fixed during fine-tunning? Or it also updating along the fine-tuning?
2) For each task, are the fine-tunning process only involved with a few-examples for each task? Or it is like traditional few-shot learning need a bunch of data like base-dataset.
3) Can the method be compared with traditional trained few-shot learning? (train a vit from scratch using the data from seen classes of its own domain). If yes, would be nice to see some numbers. If not, please further explain a little more.
4) When utilizing the pre-trained transformer, how does 5-way few shot classification works? Would be nice to see some numbers for comparison. I am wondering if it is too easy or too little data to be fine-tunned.
5) For the compared baselines, are the MetaOptNet and SVM the same for fixed backbone? I am wondering why they differ so much in terms of the performance.

If the questions are addressed, I am willing to raise the score. Thank you in advance!

**Limitations:**

In general, I think the contribution might not be enough since the pre-trained model are proofed very powerful already. If the questions are addressed, I am willing to raise the score. Thank you in advance!

---

> ### Author Rebuttal · Authors · 2023-08-09
>
> Please check the reference rules in the global response first to help following reading.
>
> **1.fixing or updating the position prompts along fine-tuning**
>
> The position prompts are obtained before fine-tuning and fixed as supervision signals during fine-tuning process. We find that updating them along fine-tuning can not obtain non-trivial improvement but introduces obvious cost.
>
> **2.only support set of each novel task or using a base-dataset**
>
> In this work, we aim to effectively adapt the pre-trained large models to target few-shot tasks. Therefore, for each task, we only have its support set that contains few labeled samples and a pre-trained model. We directly fine-tune the pre-trained large model in the few support samples, which is actually very challenging, especially for the data-hungry vision transformers.
>
> **3.comparison with traditional few-shot learning**
>
> In this work, we abandon the traditional setting that learns inductive bias from a small base dataset, and instead propose a new paradigm that directly adapts the models pre-trained on massive data to target few-shot tasks. This new paradigm has many advantages: 1) the pre-trained large models have far more generalizing representation and thus can obtain significantly better performance; 2) it can handle supervised, cross-domain and unsupervised few-shot learning simultaneously: the pre-training data can be unlabeled and thus sufficiently large, and the novel tasks can come from different target domains; 3) it is more friendly to privacy data, and we only need the pre-trained models instead of pre-training data; 4) it has better realistic value and conforms to the learning paradigm of humans whose few-shot ability comes from learning with massive signals since birth. The reason why we use the 20-way setting is also to make the setting more realistic and practical useful. The 5-way setting is too simple to be a valuable one.
> Most traditional few-shot learning methods typically design some parametric meta modules and need a base dataset for meta-training, and thus are not applicable in our setting. The results of some applicable ones, e.g., MetaOptNet and R2D2, are compared in Tab.1,2,3,4-main.
>
> **4.results on 5-way 1-shot/5-shot tasks**
>
> We provide the results on 5-way tasks in Tab.1-rebuttal. The performance are far more better than traditional few-shot learning methods. For the 1-shot tasks from miniImageNet, we can achieve 94.2% accuracy while the SOTA traditional method, FeLMi[a], only achieve 68.28%. For the 1-shot tasks from CUB, we can achieve 84.4% accuracy while the SOTA traditional method, Wave-SAN[b], only achieves 50.3%. This means using pre-trained large models is more effective and practically valuable than traditional paradigm. Moreover, our method can further improve the fine-tuning performance of base fine-tuning methods, e.g., LoRA and SSF.
>
> **5.difference between MetaOptNet and “SVM”**
>
> For “SVM”, we use the method from scikit-learn library, i.e.,
>
>     from sklearn.svm import SVC
>     classifier = SVC(C=1., kernel='rbf')
>
> whose implementation is based on libsvm library. Differently, MetaOptNet use multi-class kernel-based vector machines from [c]. Both MetaOptNet and “SVM” use the same fixed backbone, but the different multi-class support vector machine methods lead to different performance.
>
> **6.novelty of denoising**
>
> Denoising for better explanation is indeed a commonly-used method, e.g., Eigen-CAM[d], and we do not treat it as a contribution and thus do not discuss about denoising in the related works. Our main innovation is to effectively combine the attention and gradient information to obtain a method that can localize the key patches accurately and generally. The attention information is not enough for some pre-trained models, like CLIP, to locate the key patches, so we introduce the denoised gradient to the attention map, as shown in Eq.6-main.
>
> **7.contribution and improvement**
>
> (1)Our contribution is that we propose a new form of prompts for vision data, position prompts, to effectively adapt the pre-trained vision transformers to few-shot image classification tasks. We provide their definition, obtaining method (Section 3.4) and usage (Section 3.5). The reason why the usage is a simple regularization loss besides the classification loss is the limitation of the new setting. Concretely, for a target few-shot task, we only have few support samples without base dataset, so it is impossible to learn a new parametric module from scratch. This greatly limits the designing space of the model, and we have to resort to the simple non-parametric method. Our regularization method introduces no parametric modules and is simple, but can effectively enhance the attention of model to the key patches. (2) We conduct extensive experiments for different backbone, pre-training ways, fine-tuning methods and datasets. Our method can obtain significant improvement (+2.0~6.0%) in some cases, and also obtains small improvement in other cases. We believe this is normal in deep learning, as it is impossible for a method to be significantly effective in all cases.
>
> Finally, although the pre-trained models are proofed very powerful already, it is still meaningful to effectively adapt them to few-shot tasks to obtain better performance. In fact, it is a non-trivial problem, since fine-tuning the large data-hungry vision transformers on few samples is prone to over-fitting. As shown in Tab.1-main, the fine-tuning methods sometimes cannot significantly improve performance beyond the initial classifier (i.e., MetaOptNet) or even perform worse, especially on 1-shot tasks.
>
> [a]FeLMi: few shot learning with hard mixup. NeurIPS 2022.
> [b]Wave-SAN: Wavelet based Style Augmentation Network for Cross-Domain Few-Shot Learning. arXiv:2203.07656.
> [c]On the algorithmic implementation of multiclass kernel-based vector machines. JMLR 2001.
> [d]Eigen-CAM: Class Activation Map using Principal Components. IJCNN 2020.

---

> > ### Comment · Reviewer_SFDV · 2023-08-19
> > **Response to the rebuttals.**
> >
> > I appreciate the authors for your effort for the comprehensive rebuttals. This rebuttals addresses most of my questions. I still have concerns as follows. Traditional setting of few-shot learning can already get very good performance with a smaller backbone, for example, TDM[1] reaching 84.36 on 1-shot 5-way classification. This shows that with a much smaller dataset (the base dataset) and smaller backbone, it can obtain SOTA results. It is unnecessary to utilize much more data and larger backbone to solve such easy problem. Plus, [2] has also shows that utilizing the pretrained model can easily adapt to this few shot setting. I get the point that the model make the fine-tuning a little more effective, but only testing on this easy classification task with 1% improvement is not persuasive. It would be nice to show this regularizer are effective on different few-shot downstream tasks e.g. on object detection since authors shows the good semantics in the attention maps. I will remain my score around boarderline for now and make my final decision later to see other reviewers and AC's opinion.
> >
> > [1] Task Discrepancy Maximization for Fine-grained Few-Shot Classification, CVPR 2022
> >
> > [2] Pushing the Limits of Simple Pipelines for Few-Shot Learning: External Data and Fine-Tuning Make a Difference CVPR 2022

---

> > > ### Author Response · Authors · 2023-08-20
> > > **Thanks very much for you feedback.**
> > >
> > > **1.advantage of adapting large self-supervised pre-trained model**
> > >
> > > First, the traditional setting which uses a base dataset is not inherently a true few-shot setting. If we want to obtain a few-shot model for CUB, we have to collect many labeled bird images for meta-training, which still needs massive labeling labor and seriously hinders the practical application. Importantly, for different domains, e.g., CUB, Cars, we have to label separate base datasets for them. Besides, the SOTA cross-domain accuracy to CUB is only 50.3%[a]. Our new setting uses open-source large models pre-trained on massive unlabeled data and only uses the few support samples from different target domains for fine-tuning, which is a true few-shot setting and can handle different domains simultaneously. Second, TDM[b] achieves 84.36% on 5way-1shot tasks from CUB, but only achieve 56% for Pets and we can achieve 91.2% as shown in Tab.1-rebuttal. Therefore, TDM can not obtain stable good performance. The few-shot problem is actually not a easy problem, just the old 5-way setting is too simple and not practical.
> > >
> > > **2.about improvement**
> > >
> > > First, [c] uses a pipeline, pre-training+meta-training+fine-tuning, which still needs to label massive in-domain data for different target domains. Instead, we consider a more challenging setting with only a pre-trained model and few support samples. Second, the 95% confidence interval is 0.3 (1-shot) and 0.2 (5-shot) as provided in line_269-main, so we believe the >1% improvement is meaningful enough. In fact, directly fine-tuning the large pre-trained model on few (20~100 in total) samples is a very challenging problem and we conduct the first exploration. Even [c] still needs a base dataset for meta-training. Third, we focus on few-shot classification in this work, and our method may have the potential to work on detection tasks, so we leave this for future work.
> > >
> > > [a]Wave-SAN: Wavelet based Style Augmentation Network for Cross-Domain Few-Shot Learning. arXiv:2203.07656.
> > > [b]Task Discrepancy Maximization for Fine-grained Few-Shot Classification, CVPR 2022
> > > [c]Pushing the Limits of Simple Pipelines for Few-Shot Learning: External Data and Fine-Tuning Make a Difference. CVPR 2022

---

> > > ### Author Response · Authors · 2023-08-21
> > > **Do our responses solve you concerns?**
> > >
> > > Thank you very much for your feadback and hard work. Do our responses solve you further concerns? If you have other concerns, welcome to discuss with us and we will be very happy to answer them.

---

### Official Review · Reviewer_h2aC · 2023-07-06

**Soundness:** 2 fair
**Presentation:** 2 fair
**Contribution:** 2 fair
**Rating:** 5
**Confidence:** 4

**Summary:**

This paper addresses few-shot learning by resorting to pre-trained large vision models. A new prompt strategy, termed position prompt, is proposed during fine-tuning to encourage the model to focus on class-relevant patches. This is realized by an attention-based token selection module and an optimization objective that enhances the affinity values between image tokens and the selected key patches. The proposed method is applied to various Vision Transformer structures with different fine-tuning strategies to verify the effectiveness of the proposed method. Experiments on a set of few-shot learning tasks show the proposed method can effectively boost few-shot learning performance.

**Strengths:**

1. This paper makes an interesting attempt on how to prompt large pre-trained vision models during fine-tuning and is found beneficial for few-shot learning.
2. It is good to see the proposed prompt strategy is applied to different combinations of Vision Transformer architectures and fine-tuning strategy to verify the generalizability of the proposed method.

**Weaknesses:**

1. The claim regarding the importance of locating class-specific patches appears too strong. Firstly, the datasets used in this paper primarily consist of single objects with context, so the statement that "the input images typically contain multiple entities" may not hold true. Secondly, contextual information is not always detrimental to classification and often provides valuable cues for recognizing the objects of interest.

2. While the method employs a sophisticated strategy to locate key patches using both MHA and gradient information, this does not guarantee the accuracy of key patch identification. As there are no ground-truth annotations for key patches, limited visualizations alone are insufficient to justify their correctness. If the identified patches do not align with expectations, it contradicts the original motivation.

3. The authors have overlooked existing attempts to locate discriminative image parts in few-shot learning that aim to enhance performance, such as the works referenced [1] and [2].

[1] "Multi-attention Meta Learning for Few-shot Fine-grained Image Recognition." IJCAI 2020.
[2] "Multi-attention network for one-shot learning." CVPR 2017.

**Questions:**

In Eq. 5, G is a vector, this vector is repeated before applying to Eq. 6? What is the motivation to project gradients to input to its first principle component?

**Limitations:**

Limitation analysis is provided.

---

> ### Author Rebuttal · Authors · 2023-08-09
>
> Please check the reference rules in the global response first to help following reading.
>
> **1.vector *G* in Eq.6-main**
>
> Yes, the vector *G^T* with shape of (1, *N*) is repeated row-wise to add to the attention matrix with shape of (*N*, *N*).
>
> **2.motivation to reserve the first principle component of gradient**
>
> The gradient computed for single sample contains many noises and we reserve its first principle component for denoise. The similar idea can also be seen in some deep explanation methods, i.e., Eigen-CAM[a]. In fact, for the gradient *G* with shape of (*N*, *d*) where *N* is the patch number, we find keep the maximum value of each patch gradient can achieve similar effect, i.e., max(*G*, dim=1) --> *G* with shape of (*N*, 1).
>
> **3.importance of locating class-specific patches**
>
> Firstly, although the used datasets are typically object-centric, the images still contain multiple entities in most cases, e.g., birds and tree branches, cars and buildings, etc. The class-independent objects introduce noise information and harm the performance, especially for scarce training samples. Secondly, most contextual information can not reflect the essential characteristics of target classes, and thus misleads classification. Whether a bird lands on a tree branch or a roof should not affect its class. Besides, if some contexts are useful for recognizing the objects, we also treat their corresponding patches as the key ones.
>
> **4.correctness of locating the key patches**
>
> We agree that the qualitative visualization of position prompts does have limited justification. Since there are no ground-truth annotations, it is impossible for us to provide quantitative verification. However, we conduct extensive visualization of position prompts beyond the few ones in the main text, and find our method can indeed effectively locate key patches. In fact, the supplementary materials also include some samples in Figures_ViT-B_DINO.zip, with the position prompts of 140 images, and please check them for further validation. Besides, it has been extensively validated that deep explanation methods can locate the most classification-related parts of input.
>
> **5.discussion about related works [b,c]**
>
> [b] proposes the channel and spatial attention modules to enhance the local features of each input. [c] aim to use class text tag to aid classification and propose a attention network which can generate the attention map based on text tag embedding to weight local features. Both methods need to meta-train the introduced parametric modules using a base dataset, and [c] even needs to conduct a new image-tag datasets. Therefore, they are not suitable for our setting where we only have a pre-trained backbone and the support set of the target task. Our method instead introduces no new parametric modules and does not require meta-training, thus is more flexible.
>
> [a]Eigen-CAM: Class Activation Map using Principal Components. IJCNN 2020.
> [b]Multi-attention Meta Learning for Few-shot Fine-grained Image Recognition. IJCAI 2020.
> [c]Multi-attention network for one-shot learning. CVPR 2017.

---

> > ### Comment · Reviewer_h2aC · 2023-08-19
> >
> > I appreciate the authors' response. However, their response has not fully addressed my concerns. Firstly, it is a factual observation that the authors did not acknowledge existing efforts in learning class-specific features for few-shot learning. More significantly, there are lingering uncertainties about whether the proposed method functions as claimed and expected. It remains challenging to assert with confidence that we can automatically pinpoint the positions of key entities, especially in few-shot scenarios. Notably, a recent study [r1] revealed the potential issue of supervision collapse in few-shot learning, where models tend to focus on partial information sufficient only to distinguish a few support samples but may struggle to generalize effectively. In response to this, the authors of [r1] employed self-supervised learning to acquire more versatile representations, which appears somewhat contradictory to the motivation of the reviewed method. In light of these considerations, I am inclined to maintain my rating as borderline reject.
> >
> > [r1] CrossTransformers: spatially-aware few-shot transfer, NeurIPS 2020.

---

> > > ### Author Response · Authors · 2023-08-20
> > > **Thanks very much for you feedback.**
> > >
> > > **1.acknowledge existing efforts**
> > >
> > > The discussion about related works has been provided in rebuttal text. We acknowledge their contributions and also explain the key differences. We will add this discussion in the revision.
> > >
> > > **2.credibility of locating the key patches**
> > >
> > > Since there are no ground-truth annotations, it is impossible to conduct quantitative verification. However, the 140 images with position prompts have already been provided in Figures_ViT-B_DINO.zip of the supplementary materials and we believe this is statistically valid. In fact, it has been extensively validated that deep explanation methods can locate the most classification-related parts of input. We also observe the same effectiveness of locating the key patches in more images beyond the above 140 ones.
> > >
> > > **3.contradictory motivation to [a]**
> > >
> > > The motivation of [a] does not contradict ours at all. The supervision collapse in [a] is for pre-training, i.e., supervised pre-training harms the transfer performance in downstream novel tasks. However, in this work, we claim to learn class-related information for fine-tuning on novel tasks, not pre-training. In fact, the pre-trained models used in this work, e.g., DINO, iBOT, are also self-supervised models.
> > > In a word, the upstream pre-training cannot learn too much information of old/base classes for better generalization, but the downstream fine-tuning needs to focus on learning information of novel classes for better classification.
> > >
> > > [a] CrossTransformers: spatially-aware few-shot transfer, NeurIPS 2020.

---

> > > > ### Comment · Reviewer_h2aC · 2023-08-20
> > > >
> > > > Thank you for your swift reply. Regarding the issue of **3. contradictory motivation to [a]**, it's important to clarify that the phenomenon of supervision collapse is not inherently linked to pre-training or fine-tuning, as long as the training is operated within a few-shot setting. In few-shot learning scenarios, the primary objective is to differentiate a small number of samples from a limited set of classes. Consequently, the gradient during training may emphasize "short-cut" information that is sufficient for achieving the specific few-shot task. How to mitigate the the occurrences of supervision collapse when calculating the loss and gradient thereof?

---

> > > > > ### Author Response · Authors · 2023-08-20
> > > > > **Thanks very much for the discussion.**
> > > > >
> > > > > In fact, the supervision collapse is indeed a problem for pre-training or meta-training, not for fine-tuning during test phase. This is why [a] claims to self-supervised meta-training. The intrinsic reason of supervision collapse is that, the base classes used for pre-/meta-learning do not need to be classified in test phase, we instead need to classify novel classes during test phase and these classes are out-of-domain relative to the base classes. Therefore, if we over-learn the base classes, it will over-fits the label information from base classes, which is just supervision collapse. Conversely, during fine-tuning, the fine-tuning classes are exactly used for final classification, so we need to learn more information from them.
> > > > >
> > > > > Besides, the “short-cut” problem is that the model uses class-independent shortcut information (i.e., texture, background) to complete the learning tasks, but does not learn valuable information for classification. However, focusing the key class-related objects is not a shortcut problem since these key objects are just the most valuable information. In fact, our attention enhancement loss can effectively avoid short-cut problem, which makes the models completing the classification tasks based on class-related information instead of short-cut information.
> > > > >
> > > > > [a] CrossTransformers: spatially-aware few-shot transfer, NeurIPS 2020.

---

> > > > > > ### Comment · Reviewer_h2aC · 2023-08-20
> > > > > > **Thank you for clarifying the contribution**
> > > > > >
> > > > > > The authors have effectively elucidated the distinctions and contributions of their method, which has successfully convinced me. I am happy to accept the paper now.

---

> > > > > > > ### Author Response · Authors · 2023-08-20
> > > > > > > **Thank you very much for your approval**
> > > > > > >
> > > > > > > We greatly appreciate your precious time and valuable discussion. Thanks!

---

### Official Review · Reviewer_TpQg · 2023-07-06

**Soundness:** 3 good
**Presentation:** 3 good
**Contribution:** 3 good
**Rating:** 5
**Confidence:** 3

**Summary:**

- The paper adapts pre-trained vision transformers for few-shot classification.
- Full/parameter-efficient fine-tuning using only a few examples may harm performance due to spurious correlations.
- The proposed method uses an additional auxiliary loss to guide the attention of the top layer to focus on the class-related patches.
- The target for this loss (“position prompts“) are obtained as a combination of deep explanation method (e.g. Rollout) and gradient information (of class logit wrt top layer input features).
- Theoretical analysis shows how the proposed auxiliary loss increases mutual information between the input tokens and key class-related patches.
- Experiments on CUB, Cars, Places, Plantae datasets show improvements over full/parameter-efficient fine-tuning methods and meta-learning based methods.

**Strengths:**

- The paper is written well and is easy to follow.

- The proposed Attention+Grad target gives better prompts than Attention alone (Fig3). (It would be good to see that it gives better prompts than Grad alone as well, or whether Grad is enough to get good position prompts.)

- Qualitative results of the resulting attention (Fig4) demonstrates improvements visually.

- Ablation wrt key hyper-params help understand their effect in different settings. (Please see Weaknesses for missing ablations)

- Theoretical analysis provides additional justification of the proposed approach.

**Weaknesses:**

- The most commonly studied settings in related work are 5-way 5-shot and 5-way 1-shot. It would be valuable to show results in these settings in order to compare with more recent methods than the ones in the paper. [a, b, c]

    - The used 20-way setting is more challenging, though it restricts comparisons with recent work

- Recent methods [a, b, c] also show results on the larger miniImageNet and tieredImageNet few-shot benchmarks

- Current SOTA pre-training methods are based on MAE. How does this method work when fine-tuning MAE pre-trained ViT?

- Cross-domain results to demonstrate that the method generalizes to unseen domains (as in [59] whose experimental setup is used)

- Ablation experiment using only the gradient term in Eq 6 to show the relative contribution of A

- Missing comparisons and mention of closely related work [d, e]

[a]: Roy, Aniket, et al. "FeLMi: few shot learning with hard mixup." Advances in Neural Information Processing Systems 35 (2022)
[b]: Yiren Jian, et al. "Label hallucination for few-shot classification." In Proceedings of the AAAI Conference on Artificial Intelligence (2022)
[c]: Afrasiyabi, Arman, Jean-François Lalonde, and Christian Gagné. "Associative alignment for few-shot image classification." Computer Vision–ECCV 2020
[d]: Hou, Ruibing, et al. "Cross attention network for few-shot classification." Advances in neural information processing systems 32 (2019)
[e]: Jiang, Zihang, et al. "Few-shot classification via adaptive attention." arXiv preprint arXiv:2008.02465 (2020)

**Questions:**

- DINO attention in Fig3 looks much better than Orig attention in Fig4. Is there a difference in how they are obtained?

- How do the position prompts look when only using gradient term in Eq6 ?

- Above points from the Weaknesses section

**Limitations:**

- The authors have discussed limitations:
    - The method is not applicable for non-attention based network architectures

    - Obtaining the position prompts may need manual labeling from experts for domains like medical, satellite, etc

---

> ### Author Rebuttal · Authors · 2023-08-09
>
> Please check the reference rules in the global response first to help following reading.
>
> **1.difference between the attention visualization in Fig.3-main and Fig.4-main**
>
> Fig.3-main visualizes the attention scores where the different colors represent different values, while Fig.4-main chooses the patches with top highest attention scores and covering about 95% attention for visualization where the chosen patches have the same color (i.e., white) no matter the attention scores. This can more clearly show where the models pay most attention to. Besides, the attention score visualization corresponding to Fig.4-main is provided in Fig.1-appendix.
>
> **2.position prompts using only gradient information**
>
> We follow the setup of Fig.3-main to visualize the position prompts using only gradient information. The results are shown in Fig.3-rebuttal as “Grad Prompts” and the other ones are the same as Fig.3-main. As we can see, using only gradient information also can effectively locate the position prompts, but there could be some bias due to gradient noise.
>
> **3.results on 5-way 1-shot/5-shot tasks and comparison with [a,b,c]**
>
> In this work, we abandon the traditional setting that learns inductive bias from a small base dataset, and instead propose a new paradigm that adapts the models pre-trained on massive data to target few-shot tasks. The new paradigm has many advantages: 1) the pre-trained large models have far more generalizing representation and thus can obtain significantly better performance; 2) it can handle supervised, cross-domain and unsupervised few-shot learning simultaneously: the pre-training data can be unlabeled and thus sufficiently large, and the novel tasks can come from different target domains; 3) it is more friendly to privacy data, and we only need the pre-trained models instead of pre-training data; 4) it conforms to the learning paradigm of humans whose few-shot ability comes from learning on massive signals since birth, and has better realistic value. Besides, we use the 20-way setting since it is more challenging and practically useful, and the 5-way setting is too simple.
> We provide the results on 5-way tasks in Tab.1-rebuttal. The results are far more better than those in [a,b,c], e.g., for 1-shot tasks from miniImageNet, the best accuracy of [a,b,c] is 68.28% and we can achieve 94.2%; for 1-shot tasks from CUB, [a] achieves 51.66% and we achieve 84.4%. This means using pre-trained large models is more effective and practically valuable than traditional paradigm [a,b,c]. Our FORT can also improve the performance of fine-tuning methods on 5-way tasks. Besides, [a,b,c] aim to pseudo-label the base dataset as auxiliary data, and this is not suitable for our setting. First, we may not obtain the pre-training data due to privacy. Second, even the pre-training data can be obtained, it is still too cost to pseudo-label the massive data, not like the small base dataset.
>
> **4.fine-tuning MAE pre-trained models**
>
> Although the MAE pre-trained models can obtain SOTA fine-tuning performance in dense prediction tasks, e.g., detection and segmentation, their can not obtain highly linearly discriminant representation like DINO and are not suitable for few-shot image classification. For example, fine-tuning ViT-B/16 from MAE with LoRA can only obtain 12.4% and 51.2% accuracy on 20-way 1-shot and 5-shot tasks from CUB respectively, far lag behind DINO pre-trained models whose results are 57.9% and 88.2% respectively.
>
> **5.cross-domain results**
>
> Our new setting is essentially a cross-domain setting: the massive pre-training data can come from different domains with target few-shot tasks, and even be unknown if there are privacy issues. We can adapt the pre-trained models to the few-shot tasks from any target domains.
>
> **6.ablation experiment using only the gradient term**
>
> We follow the setup of Section_4.1-main and ablate the attention and gradient information. The results are shown in Tab.2-rebuttal where “+FORT (grad)” denotes using only gradient term, “+FORT (attn score)” denotes directly using attention scores, “+FORT” denotes using both attention and gradient information. Here we pick the best hyper-parameters for each model even though they may differ from each other. For DINO pre-trained models, the internal attention can locate the key patches and thus effectively assist the gradient information. For CLIP pre-trained models, the internal attention can not indicate key patches and the gradient information plays a major role.
>
> **7.comparison with related work [d,e]**
>
> [d] aims to use the correlation between the local features of support and query samples to calculate similarity precisely and proposes a cross-attention module. [e] aims to highlight the related features in query samples based on support samples and proposes the meta weight generator and spatial attention generator. These models follow the traditional meta-learning paradigm and need a base dataset to meta-learn the parametric module, thus can not applied in our new setting. Our method introduces no new parametric modules and does not require meta-training, thus is more flexible.
>
> [a]FeLMi: few shot learning with hard mixup. NeurIPS 2022.
> [b]Label hallucination for few-shot classification. AAAI 2022.
> [c]Associative alignment for few-shot image classification. ECCV 2020.
> [d]Cross attention network for few-shot classification. NeurIPS 2019.
> [e]Few-shot classification via adaptive attention. arXiv:2008.02465.

---

### Official Review · Reviewer_8eth · 2023-07-10

**Soundness:** 2 fair
**Presentation:** 3 good
**Contribution:** 2 fair
**Rating:** 5
**Confidence:** 4

**Summary:**

This paper proposes to force the pre-trained model to focus on class-related entities for few-shot image classification. To achieve this, it proposes position prompts that use attention and gradient information to automatically locate the positions of key entities. An attention enhancement loss is used to be trained with the cross-entropy loss. Experiments are conducted on CUB, Cars, Places, Plantae, Aircraft, and Pets datasets.

**Strengths:**


+ The idea to focus attention for few-shot learning is quite interesting.
+ This paper is well-organized, with intuitive figures which illustrate the motivation and approach clearly.
+ The experiments are extensive.


**Weaknesses:**

+ The statement in Ln168 “When the support samples are sufficient, the model can attend to the frequently-occurring entities in each class more to alleviate this problem, since they are typically the key ones” can’t be well-supported by Fig. 4. Instead, attention maps with **sufficient training samples** should be included to prove that the “attention focus” is especially useful for few-shot learning.
+ More illustration of the hyper-parameters $\alpha$ should be included as it controls the core component of the proposed approach, the attention enhancement loss. For example, why does the performance decrease with higher $\alpha$; why is the performance of 5-shot tasks more sensitive to the value? If as stated in Ln168, more samples would be less sensitive to the attention selection?
+ The illustration of the position prompts is unclear and quite difficult to understand, which makes it difficult to evaluate its effectiveness.


**Questions:**

Please see the weaknesses.

**Limitations:**

Yes.

---

> ### Author Rebuttal · Authors · 2023-08-09
>
> Please check the reference rules in the global response first to help following reading.
>
> **1.attention maps with sufficient training samples**
>
> We follow the setup of Fig.4-main, except fine-tuning on the 20-way 50-shot task from CUB and 20-way 200-shot task from Pets without attention enhancement. The attention maps after fine-tuning are shown in Fig.1-rebuttal and Fig.2-rebuttal as “FT Attn (sufficient)”. As we can see, fine-tuning with sufficient samples indeed focuses on the key entities more than with few samples (“FT Attn”) or original attention (“Orig Attn”), but is less focused than explicit attention enhancement (“Our Attn”).
> Note that this does not mean that few-shot fine-tuning with attention enhancement can obtain higher accuracy than sufficient-shot fine-tuning. In fact, sufficient training samples allow the models to fit the input-label distribution better, which is more important for classification than attention enhancement. But when we can not obtain sufficient samples, attention enhancement can reduce class-independent noise information in the final embedding as shown experimentally and theoretically, and thus is useful for classification.
>
> **2.more explanation about hyper-parameter alpha**
>
> As an auxiliary of classification loss, the attention enhancement loss aims to guide the model to learn to classify the images by focusing on their key entities, so as to ignore class-independent information. Both the attention enhancement loss and the classification loss help to get better classification boundaries, but the latter one fits the input-label distribution and thus is more direct and effective. The attention enhancement loss aims to reduce noise when data is scarce and indirectly improves the generalization performance of the classifier. Therefore,
> (1) If the coefficient alpha is too large, the attention enhancement loss could relatively inhibit the optimization of the classification loss (e.g., skew gradient direction), which is harmful to fitting input-label distribution.
> (2) The more training samples, the more important it is to fit input-label distribution, so 5-shot tasks prefer smaller alpha than 1-shot tasks.
>
> **3.about position prompts**
>
> For each input image, there are some patches corresponding to the key class entity, denoted as key patches. Their positions are used as “position prompts” to guide fine-tuning, i.e., the red patches in Fig.3-main.
> As described in Section_3.4-main, we use both attention and gradient information to locate position prompts for generality. In fact, it is difficult to locate them using only attention information sometimes, as shown in Fig.3-main. “Attention+Grad Prompts” is generally better than “Attention Prompts”.
> As described in Section_3.5-main, we use the position prompts as the prediction target for attention module, and optimize the cross-entropy loss between their many-hot presentation and the attention logits during fine-tuning, i.e., Eq.8-main. Unlike the existing prompts in NLP[a], our position prompts are not used in the input or middle phase of forward inference, since those of query samples are unknown without label information.
> Finally, their effectiveness is quantitatively verified in Tab.1,2,3,4-main and qualitatively verified in Fig.4-main. The attention enhancement using position prompts indeed helps the model to focus most attention on key entities.
>
> [a] pre-train, prompt, and predict: A systematic survey of prompting methods in natural language processing.

---

> > ### Comment · Reviewer_8eth · 2023-08-21
> > **Thanks for the rebuttal**
> >
> > I appreciate the rebuttal and the clarification. The responses address most of my concerns. Therefore, I would retain my score.

---

### Author Rebuttal · Authors · 2023-08-09

We really appreciate all the reviewers for taking your precious time to make the valuable comments. Overall, **our work has the following strengths**:
1. **the proposed new setting is more realistic and has great value. [nhXH]**
2. **reasonable and interesting idea. [8eth], [SFDV], [nhXH]**
3. **sensible and useful theoretical analysis. [TpQg], [SFDV]**
4. **well-written and easy to follow. [8eth], [TpQg], [SFDV]**
5. **extensive experimental results. [8eth], [h2aC], [SFDV], [nhXH]**

We will carefully respond to the concerns of each reviewer. The reference contents come from the main text, the appendix or the rebuttal PDF. Therefore, for the convenience of reference, we mark "**xxx-main**", "**xxx-appendix**" and "**xxx-rebuttal**" to be the corresponding content xxx in the main text, the appendix and the rebuttal PDF respectively. For example, "Tab.1-main" refers to the Table 1 in the main text, and "Fig.1-rebuttal" refers to the Figure 1 in the rebuttal PDF. The valuable suggestions from reviewers will be added to the revision, such as more ablation results, more discussions about related works and more analysis on observations.

---

### Author Response · Authors · 2023-08-18
**Looking forward to further discussion**

We would like to express our sincere gratitude to all area chairs and reviewers for your precious time and hard effort.
Your comments are truely valuble for improving our work and we have provided detailed responds to address each of your concerns.
Please check our responds and re-evaluate our work. Thank you very much.
If you have some other concerns, welcome to discuss with us and we will be very happy to answer them.

---

### Decision · Program_Chairs · 2023-09-21

**Decision:**

Accept (poster)

**Comment:**

This paper obtained 4 borderline accepts and 1 borderline reject. All reviewers appreciated the interesting and intuitive idea, its practical value, and the extensive experiments, but there were concerns about unclear justification, missing related work, and limited analyses/comparisons. The authors addressed most of them in the rebuttal and subsequent discussion, providing additional experiments/analyses that show consistent improvement by the proposed technique. After carefully checking the reviews, the rebuttal, and the discussion, AC finds that the raised concerns are resolved and this submission has a good contribution to the few-shot learning area, recommending the acceptance.